# Immune checkpoint modulation enhances HIV-1 antibody induction

Todd Bradley[1,2,3,4✉], Masayuki Kuraoka[5], Chen-Hao Yeh [5], Ming Tian[6], Huan Chen[6], Derek W. Cain[1,2], Xuejun Chen [7], Cheng Cheng[7], Ali H. Ellebedy[8,14], Robert Parks[1], Maggie Barr[1], Laura L. Sutherland[1], Richard M. Scearce[1], Cindy M. Bowman[1], Hilary Bouton-Verville[1], Sampa Santra[9], Kevin Wiehe[1,2], Mark G. Lewis[10], Ane Ogbe [11], Persephone Borrow[11], David Montefiori[1,12], Mattia Bonsignori[1,2], M. Anthony Moody[1,5,13], Laurent Verkoczy[1,15], Kevin O. Saunders [1,12], Rafi Ahmed [8], John R. Mascola[7], Garnett Kelsoe[1,5], Frederick W. Alt [6] & Barton F. Haynes[1,2,5✉]

Eliciting protective titers of HIV-1 broadly neutralizing antibodies (bnAbs) is a goal of HIV-1 vaccine development, but current vaccine strategies have yet to induce bnAbs in humans. Many bnAbs isolated from HIV-1-infected individuals are encoded by immunoglobulin gene rearrangments with infrequent naive B cell precursors and with unusual genetic features that may be subject to host regulatory control. Here, we administer antibodies targeting immune cell regulatory receptors CTLA-4, PD-1 or OX40 along with HIV envelope (Env) vaccines to rhesus macaques and bnAb immunoglobulin knock-in (KI) mice expressing diverse precursors of CD4 binding site HIV-1 bnAbs. CTLA-4 blockade augments HIV-1 Env antibody responses in macaques, and in a bnAb-precursor mouse model, CTLA-4 blocking or OX40 agonist antibodies increase germinal center B and T follicular helper cells and plasma neutralizing antibodies. Thus, modulation of CTLA-4 or OX40 immune checkpoints during vaccination can promote germinal center activity and enhance HIV-1 Env antibody responses.

[1] Duke Human Vaccine Institute, Duke University School of Medicine, Durham, NC 27710, USA. [2] Department of Medicine, Duke University School of Medicine, Durham, NC 27710, USA. [3] Center for Pediatric Genomic Medicine, Children's Mercy Kansas City, Kansas City, MO 64108, USA. [4] Department of Pediatrics, UMKC School of Medicine, Kansas City, MO 64108, USA. [5] Department of Immunology, Duke University School of Medicine, Durham, NC 27710, USA. [6] Program in Cellular and Molecular Medicine, Boston Children's Hospital, Department of Genetic, Harvard Medical School, Howard Hughes Medical Institute, Boston, MA 02115, USA. [7] Vaccine Research Center, National Institute of Allergy and Infectious Diseases, Bethesda, MD 20892, USA. [8] Emory Vaccine Center, Emory University, Atlanta, GA 30317, USA. [9] Beth Israel Deaconess Medical Center, Harvard Medical School, Boston, MA 02115, USA. [10] BIOQUAL, Inc, Rockville, MD 20850, USA. [11] Nuffield Department of Clinical Medicine, University of Oxford, Oxford OX3 7FZ, UK. [12] Department of Surgery, Duke University, Durham, NC 27710, USA. [13] Department of Pediatrics, Duke University School of Medicine, Durham, NC 27710, USA. [14]Present address: Division of Immunobiology, Department of Pathology and Immunology, Washington University School of Medicine, St. Louis 63110, USA. [15]Present address: San Diego Biomedical Research Institute, San Diego, CA 92121, USA. ✉email: tcbradley@cmh.edu; Barton.haynes@duke.edu

HIV-1 vaccine development has focused on the initiation of B cell lineages capable of maturing to broadly neutralizing antibody (bnAb) cells. Sites of vulnerability on HIV-1 Env have been identified with bnAbs isolated from HIV-1 infected individuals[1–3] and passively administered bnAbs have prevented the infection of rhesus macaques by challenges with chimeric simian-human immunodeficiency viruses (SHIVs)[4–7]. Subsets of bnAbs have long third heavy chain complementarity regions (CDRH3s) associated with poly- or autoreactive B cell receptors (BCRs) and can predispose B cells to immune tolerance control[8,9]. Moreover, in addition to the HIV Env polypeptide backbone, many bnAbs also interact with Env host-derived glycans or viron lipids[9–12]. Thus, the maturation of B-cell lineages towards bnAb activity can be hindered by immune tolerance controls[13,14].

Despite much effort, to date no candidate vaccine has induced bnAbs in humans, yet up to 50% of HIV-1-infected individuals develop some level of plasma bnAbs[15]. Comparative analysis of cohorts of HIV-1 infected individuals who make bnAbs vs. those who do not have demonstrated subjects who make bnAbs during HIV-1 infection have higher levels of CD4+ T follicular helper (Tfh) cells[16,17], lower levels of T regulatory cells, and a higher frequency of auto-antibodies compared to those who do not make bnAbs[17]. Recently, we demonstrated the association of dysfunctional NK cells with the ability to produce bnAbs in the setting of HIV-1 infection[18]. Thus, immunoregulatory abnormalities are associated with bnAb development in HIV-1-infected individuals, suggesting that transiently recreating the HIV-1-modified immunoregulatory milieu at the time of vaccination may be crucial to the induction of clonal lineages of bnAb B cells with full neutralization breadth[13,14,17].

Immune checkpoint receptors can either inhibit or enhance immune response pathways[19]. Blocking inhibitory checkpoints has proved useful in boosting the activity of CD8+ cytolytic T-cell activity in tumor immunotherapy[20–22]. Cytotoxic T lymphocyte antigen 4 (CTLA-4) and programmed cell death 1 (PD-1) are inhibitory receptors critical for maintaining self-tolerance and for modulating the magnitude of immune responses[23–25].

In parallel with blocking inhibitory molecules, activating stimulatory receptors is also being explored for tumor immunotherapy. Tumor necrosis factor receptor superfamily member 4 (OX40) receptor is a stimulatory molecule on activated T cells and can result in enhanced activity and clonal expansion[26]. Administration of blocking (CTLA4, PD-1) or agonist (OX40) antibodies against these checkpoint regulators improves anti-tumor immune responses in animal models, and CTLA-4 and PD-1 antibodies have become approved immunotherapies for multiple tumor types in humans[19–22,27–29].

CTLA-4, PD-1, and OX40 are expressed on both regulatory and effector T cell populations that are critical for controlling B-cell responses. Studies of CTLA-4 knock-out mice show that CTLA-4 constrains CD4 Tfh numbers and controls T follicular regulatory (Tfr) cell numbers, and absence of CTLA-4 is linked to heightened B cell responses due to increased T cell help and defective regulatory supression[30]. Overall, humoral responses become enhanced in the absence of CTLA-4[30]. However, studies in which PD-1 interaction with its ligands has been disrupted by gene deficiency or antibody blockade have given more complex results, with enhancement of Tfh responses and humoral immunity being observed in some cases[31], but negative effects observed in others, perhaps due to the role of PD-1 in control of CD4 Tfr numbers and suppressive capacity[32]. In contrast, OX40 disruption results in reduced Tfh function, therefore therapeutic stimulation of this receptor should improve Tfh and B cell antibody responses[33].

Immune checkpoint modulation as an HIV-1 vaccine adjuvant to enhance HIV-1 Env B cell responses has not been evaluated. If immunization in the context of immunoregulators significantly improves germinal center and antibody responses, the brief use of checkpoint inhibitors may benefit HIV-1 vaccination strategies where the activation and expansion of rare bnAb precursors are necessary to achieve the requisite antibody titers needed for protection[34,35]. Here, we administered CTLA-4 or PD-1 blocking antibodies with a HIV-1 envelope (Env) immunogen in macaques and CTLA-4 blocking and OX40 agonist antibodies with an Env immunogen in a bnAb knock-in (KI) mice that express precursors of the CD4 binding site-directed bnAb VRC01. We demonstrate that either CTLA-4 blockade or agonistic OX40 antibody administration at the time of immunization enhanced the elicited Env antibody responses, providing proof-of-concept for transient use of these checkpoint agents as adjuvant in vaccination to enhance B-cell germinal center responses to HIV-1 Env vaccines.

## Results

**CTLA-4 blockade augments HIV-1 antibody responses in monkeys.** Cynomolgus macaque immune cells have previously been shown to bind ipilimumab (anti-CTLA-4) and nivolumab (anti-PD-1), and pre-clinical studies with these drugs have been performed in this animal model[36,37]. We immunized cynomolgus macaques ($n = 16$) with a sequential HIV-1 vaccine comprising Env proteins derived from a HIV-1-infected individual (CH505) who developed CD4 binding site (CD4 bs)-directed bnAbs (*Macaque study no. 1*, Fig. 1a)[38]. Macaques were immunized at weeks 0, 4, 8, and 12 with CH505 TF, wk 53, wk 78 and wk 100 Env gp120 recombinant proteins formulated in GLA-SE adjuvant. Monkeys were separated into 4 groups ($n = 4$/group) that received co-administration of (i) anti-CTLA-4 mAb (ipilimumab) at weeks 0, 4 and 8; (ii) anti-PD-1 mAb (nivolumab) at weeks 0 and 4; (iii) CTLA-4 + PD-1 at weeks 0 and 4; or (iv) control antibody (CH65) at weeks 0, 4, and 8 (Fig. 1a).

At week six, two weeks after the second Env immunization plus antibody co-administration, CTLA-4 Ab or CTLA-4 + PD-1 Ab treated monkeys had higher average plasma antibody binding titers (11.8 and 12.3 average log area under the curve titers [AUC], respectively) to the CH505 TF gp120 Env while PD-1Ab-treated animals had lower average binding antibody titers (9.6 average log AUC) compared with control antibody-treated animals (10.9 average log AUC) (Fig. 1b). Trends (NS; Wilcoxon–Mann–Whitney) for higher average binding to gp120 for CTLA-4 or CTLA-4 + PD-1 Ab groups were also observed after 3 immunizations (week 10) but not after 4 immunizations (week 14) that occurred without coadministration of CTLA-4 Ab, while lower average serum Env binding antibodies for PD-1 Ab-treated animals were observed after both the 3rd and 4th immunizations even after discontinuation of Ab coadministration (weeks 10 and 14, respectively; Supplementary Fig. 1A). Difficult-to-neutralize (tier-2) virus neutralization by plasma antibodies was not detected in any immunized group throughout the study, but similar to binding antibody titers, animals treated with CTLA-4 or CTLA-4 + PD-1 Abs had higher average titers of neutralizing antibodies against the tier-1 autologous CH505w4.3 virus with greater than 3-fold increase of average ID50 titers of neutralization compared to control animals, whereas those treated with PD-1 Ab had equivalent (<2-fold difference) neutralizing antibody titers, relative to the control-treated group (Fig. 1c). Monkeys treated with the combination of CTLA-4 + PD-1 Abs exhibited significantly higher plasma neutralization compared with control or PD-1 Ab treated animals ($P = 0.03$, Wilcoxon–Mann–Whitney). Trends for higher average (NS; Wilcoxon-Mann-Whitney) plasma neutralizing antibodies in the CTLA-4 and statistically significant higher titers ($P = 0.03$,

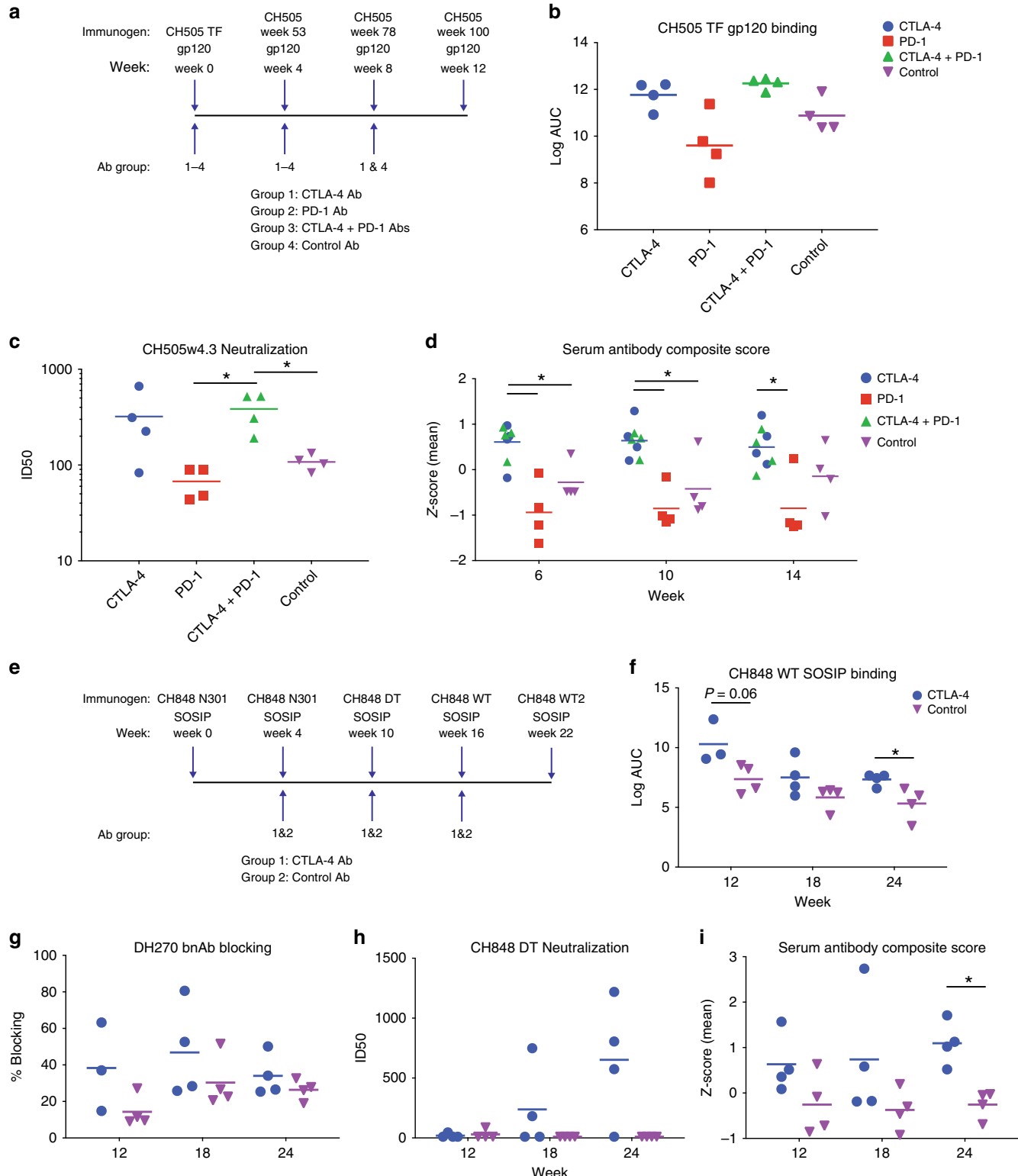

Wilcoxon–Mann–Whitney) in the CTLA-4 + PD-1 Ab treated groups were also observed at weeks 10 and 14 (Supplementary Fig. 1B).

In addition to increase in overall binding and tier 1 virus neutralizing antibodies, we assayed for plasma antibodies that could block sCD4 binding to the CH505 TF Env gp120 in a competitive ELISA assay to quantify antibodies that target the CD4 binding site (Supplementary Fig. 1C). We observed that CTLA-4 and CTLA-4 + PD-1 Ab treated animals had trends for

higher average titers of antibodies that bound to the CD4bs as measured by higher average blocking of sCD4 binding after the 2nd 3rd, and 4th immunizations, whereas macaques treated with PD-1 Ab alone-treated macaques had lower average levels of CD4 blocking antibodies (Supplementary Fig. 1C). When plasma antibody binding, CD4 binding site blocking and tier-1 neutralization measurements were converted to Z-scores and the Z-scores from each assay were averaged for each animal as a composite antibody score, CTLA-4 plus CTLA-4 + PD-1

**Fig. 1 Co-administration of CTLA-4 and PD-1 blocking antibodies with HIV vaccines in Cynomolgus macaques. a** Vaccination protocol for immunization of four groups of Cynomolgus macaques with sequential CH505 HIV Env gp120 recombinant proteins with co-administration of CTLA-4, PD-1, CTLA-4 + PD-1 or control antibodies. **b**, **c** Plasma antibody titers of (**b**) binding to CH505 TF gp120 by ELISA measured by Log area under the curve (**c**) neutralization of tier-1 autologous CH505w4.3 virus in the TZM-bl assay measured by ID50. Each dot represents a single animal and are colored based on antibody treatment group (CTLA-4, blue; PD-1, red; CTLA-4 + PD-1, green; Control CH65, purple). **d** Antibody composite score averaging the Z-scores from the binding, blocking and neutralization values measured at weeks 6, 10, and 14 (*$P < 0.05$; Wilcoxon–Mann–Whitney). **e** Vaccination protocol for immunization of four groups of Cynomolgus macaques with sequential CH848 SOSIP trimer recombinant proteins with co-administration of CTLA-4 or control antibodies. **f–h** Plasma antibody titers of (**f**) binding to CH848 WT SOSIP trimer by ELISA measured by Log area under the curve. (**g**) blocking of the DH270 V3-targeting bnAb binding to CH848 WT gp120 protein by competitive ELISA measured as percent blocking. **h** neutralization of tier-2 autologous CH848 DT virus in the TZM-bl assay measured by ID50. **i** Antibody composite score averaging the Z-scores from the binding, blocking and neutralization values measured at weeks 12, 18, and 24 (*$P < 0.05$; Wilcoxon–Mann–Whitney). Each dot represents a single animal and are colored based on antibody treatment group (CTLA-4, blue; Control CH65, purple; *$P < 0.05$; Wilcoxon–Mann–Whitney). Source data are provided as a Source Data file.

Ab-treated monkeys had significantly better composite antibody functional scores compared with PD-1 and control Ab-treated animals after the 2nd, 3rd, and 4th immunizations (weeks 6, 10 and 14, respectively; $P < 0.05$, Wilcoxon–Mann–Whitney; Fig. 1d). Animals treated with CTLA-4 alone or CTLA-4 plus PD-1 were grouped for statistical power. In contrast, PD-1 Ab-treated monkeys had a trend for lower average antibody composite score at all three timepoints but did not meet the statistical significance (Fig. 1d).

In a parallel study in BALB/c mice (Supplementary Fig. 1D), we immunized using a similar vaccine regimen and treated mice with two doses of CTLA-4, PD-1 or control blocking antibodies specific for mouse receptors (Supplementary Fig. 1D). Similar effects of higher average binding, blocking and neutralization titers in CTLA-4 Ab treated animals and lower antibody responses in PD-1 treated animals were observed ($P \leq 0.05$; Wilcoxon–Mann–Whitney; Supplementary Fig. 1E–G).

We next tested the effect of CTLA-4 blockade in *Macaque study no. 2* (Fig. 1e) using an HIV-1 native-like SOSIP trimer that was designed to engage the unmutated common ancestor antibody (UCA) encoding B cells of a V3-glycan targeting bnAb[12]. Cynomolgus macaques ($n = 16$) were immunized twice with a near-native CH848 gp140 stabilized SOSIP trimer with the asparagine (N) 301 glycosylation site removed to enhance V3-glycan unmutated common ancestor antibody (UCA) binding, followed by sequential boosts with variants of the CH848 trimer designed to sequentially replace either N301 or V1 glycans (Fig. 1e). Monkeys were co-administered CTLA-4 or control antibodies at weeks 4, 10, and 16 with SOSIP trimers (Fig. 1e). Antibody binding to the CH848 wild-type (WT) SOSIP trimer with all of the glycans intact was measured by ELISA after 3rd, 4th, and 5th immunizations at weeks 12, 18 and 24, respectively. At each timepoint, CTLA-4 Ab-treated animals had trends for higher SOSIP trimer binding titers compared with controls, and statistically significantly higher titers at week 24 ($P = 0.03$, Wilcoxon–Mann–Whitney; Fig. 1f). Similarly, CTLA-4 Ab-treated animals had trends for higher average titers of antibodies that could block the DH270 bnAb binding to the CH848 Env at weeks 12, 18, and 24 after the 3rd, 4th, and 5th immunizations, indicating induction of antibodies that overlap with the DH270 bnAb epitope (Fig. 1g). Lastly, we tested autologous neutralization against tier-2 (difficult-to-neutralize) CH848 10.17 DT virus at weeks 12, 18, and 24 after immunization. There was no neutralization detected above an ID50 of 100 after 3 immunizations, but after the 4th immunization, 2 of 4 CTLA-4 Ab treated animals had reproducible serum tier-2 autologous HIV-1 CH848 10.17 DT neutralization (mean ID50 titer of 237.8), while no neutralization was detected in any of the animals in the control Ab treated group (Fig. 1h). After five immunizations at week 24, 3 of the 4 CTLA-4 Ab treated animals had detectable serum autologous tier-2 neutralization activity (mean ID50 titer of

652.0) (Fig. 1h). When serum antibody binding, blocking and neutralization were combined into a serum antibody composite score, animals treated with CTLA-4 Ab had significantly higher scores at week 24 compared to control or other groups and averaged higher at weeks 12 and 18 ($P < 0.05$; Wilcoxon–Mann–Whitney; Fig. 1i). Although the sample size of each macaque study precluded individual measurements from reaching a statistically significant threshold of $P \leq 0.05$ (Wilcoxon–Mann–Whitney), the trends observed for multiple measurements within two independent macaque studies suggested that during HIV-1 Env immunization in macaques, CTLA-4 blockade could increase the titers and change the quality of the neutralization response to Env by promoting induction of autologous tier-2 neutralizing antibodies.

**Measurements of HIV-1 Env-specific IgG avidity by SPR.** We observed trends and statistically significant measures for increased HIV-1 Env-specifc antibody titers by ELISA in CTLA-4 Ab-treated macaques, so we next purified plasma antibody IgG and tested if CTLA-4 or PD-1 treatment had any effects on antibody avidity using surface plasmon resonance (SPR) in two replicate experiments. In *Macaque study no. 1* we tested plasma IgG isolated two weeks after the second immunization and antibody treatment (week 6) for binding, dissociation rate and avidity score to the CH505 TF gp120 protein and a native-like CH505 TF SOSIP trimer protein (Fig. 2a, b, Supplementary Fig. 2A and C). For both proteins, the IgG binding response reflected the trends observed in the ELISA assays of higer average binding for CTLA-4 and CTLA-4 plus PD-1 Ab-treated animals and lower average binding of PD-1 Ab-treated macaques (Fig. 2a and Supplementary Fig. 2C). There were no differences in the IgG avidity scores for binding to CH505 TF gp120 protein among the different groups, but CTLA-4 and CTLA-4 plus PD-1-treated macaques had higher average avidity scores for the CH505 TF SOSIP trimer when compared to controls but did not reach statistical signicance (Fig. 2b, Supplementary Fig. 2A and C).

In *Macaque study no. 2*, we assayed IgG two weeks after the 4th immunization and 3rd Ab treatment (week 18) for binding against the CH848 DT and CH848 WT SOSIP proteins (Fig. 2c, d, Supplementary Fig. 2B and D). CTLA-4 Ab-treated animals had higer average IgG binding to both proteins compared to controls (Fig. 2c and Supplementary Fig. 2D). CTLA-4 Ab-treated animals also had higher average antibody avidity scores to the CH848 DT and CH848 WT SOSIP proteins, but did not meet statistical significance in both experimental replicates (Fig. 2d, Supplementary Fig. 2B and D). The results from both studies demonstrated trends for increased antibody avidity against native-like SOSIP trimers in CTLA-4 Ab-treated macaques, but similar to the ELISA antibody titer measurements in Fig. 1, the small sample size precluded individual measurements from reaching statistical significance.

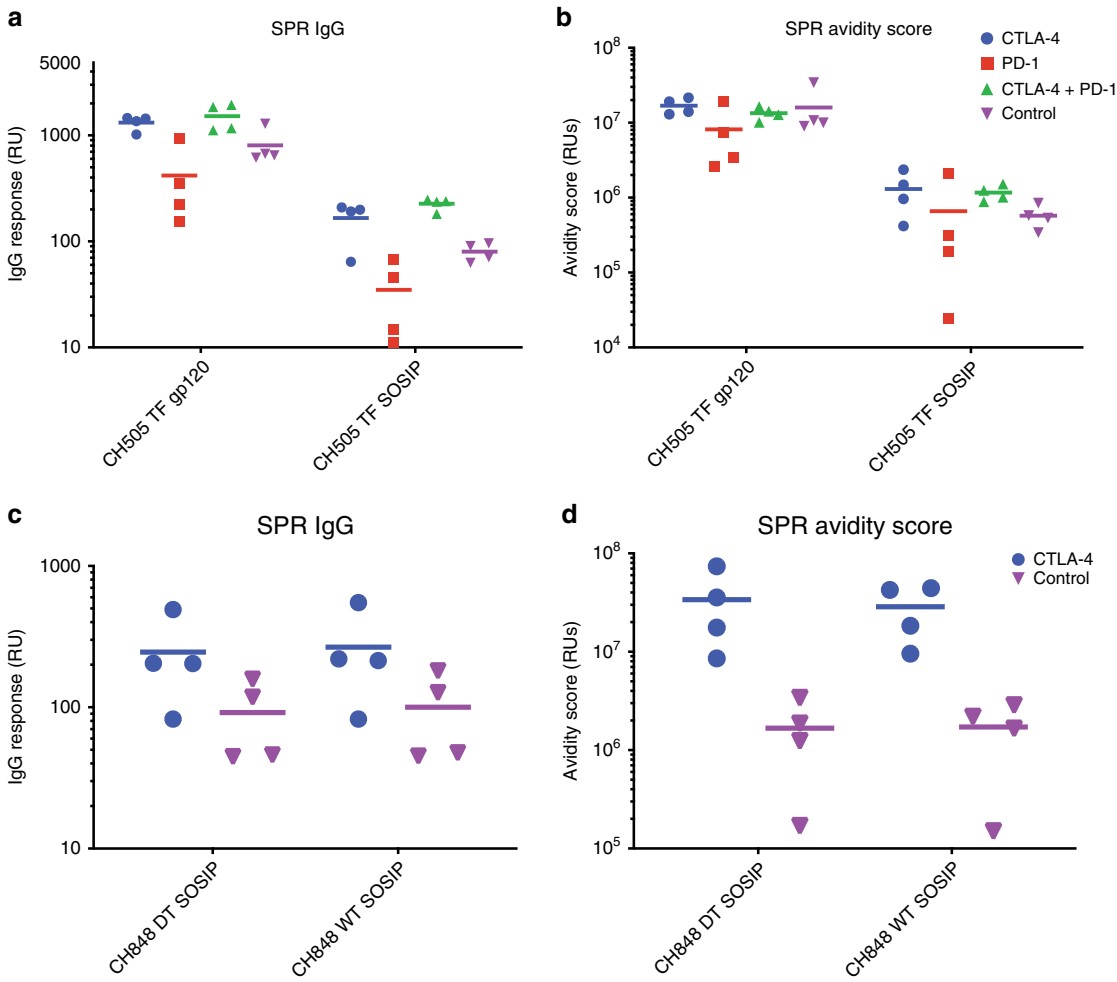

**Fig. 2 Antibody IgG affinity and avidity measured by SPR after HIV-1 Env vaccination. a**, **b** Purified plasma IgG (**a**) binding and (**b**) avidity as measured by surface plasmon resonance (SPR) to CH505 TF gp120 and CH505 TF SOSIP trimer proteins. Each dot represents a single animal and are colored based on the antibody treatment group (CTLA-4, blue; PD-1, red; CTLA-4 + PD-1, green; Control CH65, purple). **c**, **d** Purified plasma IgG (**c**) binding and (**d**) avidity as measured by surface plasmon resonance (SPR) to CH848 DT and CH848 WT SOSIP trimer proteins. Each dot represents a single animal and are colored based on the antibody treatment group (CTLA-4, blue; Control CH65, purple). Source data are provided as a Source Data file.

**Immune-checkpoint administration altered the macaque blood transcriptome of CD4+ T and CD20+ B cells.** From all animals in *Macaque study no. 1*, CD4+ T cells were isolated by flow cytometry from peripheral blood mononuclear cells (PBMC) collected at week 1 (7 days after the 1st HIV-1 Env immunization and antibody co-administration) from all monkeys in all 4 groups from animals in the first study (Fig. 1a), and RNA-seq was performed to determine the effect of checkpoint blockade on CD4+ T cell transcriptional networks (Supplementary Fig. 3A). We identified 74, 120, and 92 transcripts that were significantly different in CTLA-4, PD-1 and CTLA-4 + PD-1 Ab groups compared to control monkeys, respectively with some differentially regulated transcripts being shared among groups (Fig. 3a, b; Supplementary Data 1–3). Of 13 differentially-expressed transcripts that were common to the CTLA-4 and CTLA-4 + PD-1 Ab-treated groups, 11 were upregulated, 10 of which are involved in T cell proliferation and activation, including transcripts that encode KI67 (*MCM5*, *BIRC5*, *MKI67*, *ZBP1*, *TK1*, *NCAPG*, *MYBL2*, *NBL1*, *CDT1*, *IFI27*), plus 1 transcript involved in cell adhesion (*MYO1F*), and 2 transcripts were downregulated, which have uncharacterized functions (*FAM60A*, *ENSMMUG00000009155*; Fig. 3b). We also found other transcripts previously shown to be involved in T cell activation (*RRM2*, *E2F8*, *TLR2*, *SECTM1*, *FOXM1*) that were upregulated in either the CTLA-4 or the CTLA-4 + PD-1 Ab

groups individually, when compared to control monkeys. These transcripts were more highly expressed than in control monkeys, and were also more highly-expressed relative to the PD-1 Ab-treated animals. Moreover, PD-1 Ab-treated monkeys had upregulation of transcripts associated with decreased memory T cell responses (*SLA*) and T cell anergy (*NDRG1*)[39,40] (Supplementary Data 2). Performance of enriched pathway analysis of the transcripts changed in the PD-1 treated animals compared with controls demonstrated significant upregulation of transcripts involved with cellular inhibition of activation (Supplementary Fig. 3B). These data demonstrated that CTLA-4 Ab treatment increased transcripts associated with T cell proliferation and activation, while PD-1 Ab-treated monkeys had upregulation of transcripts associated with CD4 T cell inhibition of activation and proliferation (*SLA*, *NDRG1*, *FOXP3*, *IFNG*, *OSM*, *SEMA7A*, *SERPINE1*, *TGFB1*, *VDR*, *CSF1*, *FASLG*).

We next determined the effect of CTLA-4 and PD-1 blocking antibodies on the transcriptome networks of B cells by performing RNA-seq of flow-sorted CD14-CD3-CD20+ B cells from PBMCs isolated one week after the first immunization from all 16 monkeys in the 4 groups from *Macaque study no. 1* (Supplementary Fig. 3A). Eighteen, 38 and 35 significant differentially expressed transcripts in the CTLA-4, PD-1 and CTLA-4 + PD-1 groups, respectively, that were differentially

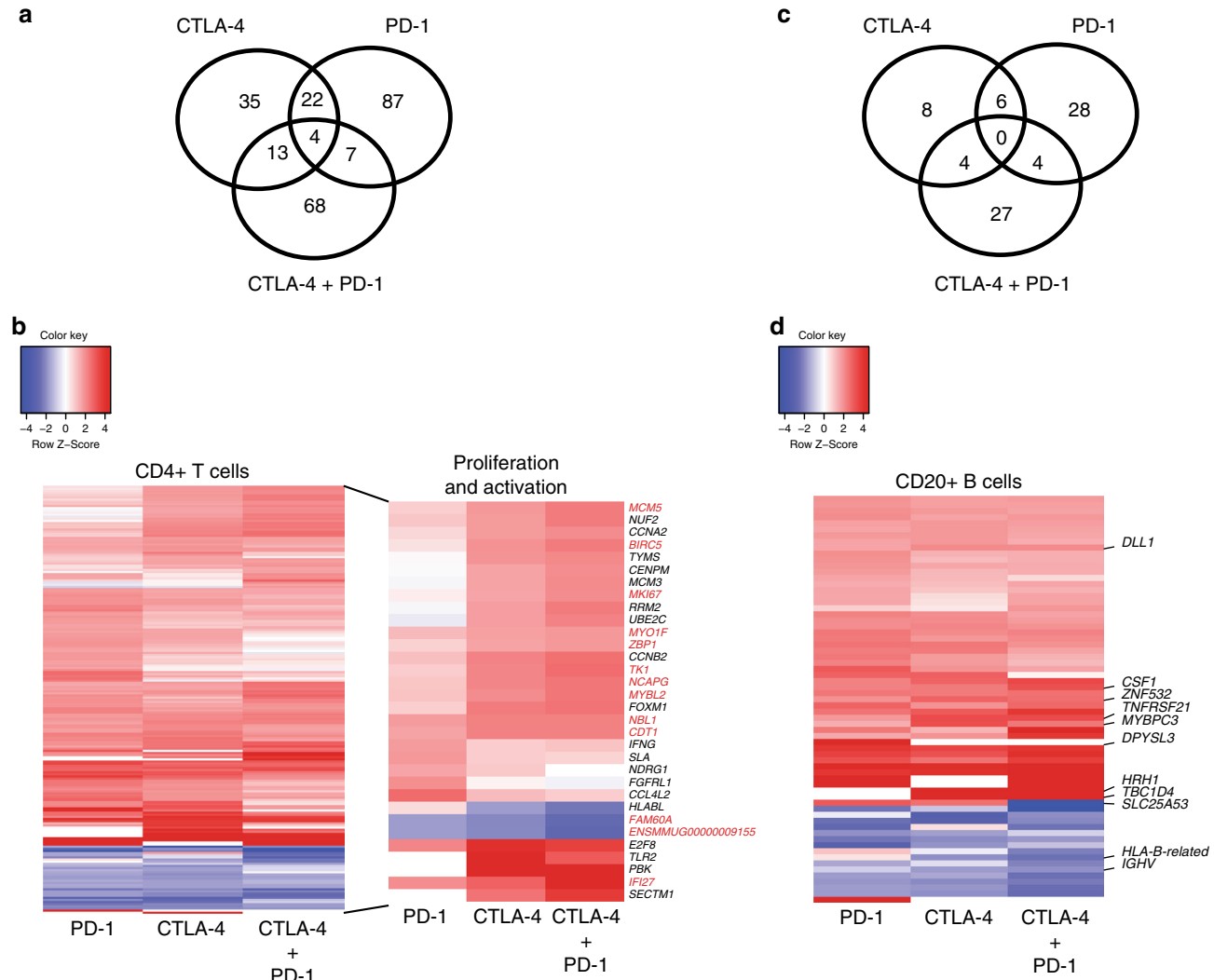

**Fig. 3 Immune checkpoint blockade alters blood immune cell transcriptome. a** Venn diagram of number of significantly changed genes compared to control group for CTL-4, PD-1, and CTLA-4 + PD-1 Ab treated animals CD4+ T cells. **b** Heatmap of genes significantly changed in PD-1, CTLA-4, and CTLA-4 + PD-1 Ab-treated animals CD4 T cells isolated from PBMCs at week 1 compared with control treated animals (left). Heatmap of a selected subset of transcripts that are involved in CD4 T cell proliferation and activation that were significantly changed in both CTLA-4 and CTLA-4 + PD-1 Ab-treated macaques (red) or in each individual experimental group and control comparison (black; right). **c** Venn diagram of number of significantly changed genes compared to control group for CTL-4, PD-1 and CTLA-4 + PD-1 Ab treated animals CD20+ B cells. **d** Heatmap of genes significantly changed in PD-1, CTLA-4 and CTLA-4 + PD-1 Ab-treated CD20+ B cells isolated from PBMCs at week 1 compared with control treated animals.

expressed compared to the control group were identified with a small number of transcripts overlapping between groups (Fig. 3c, d; Supplementary Data 4–6). Both the CTLA-4 and CTLA-4 + PD-1 groups had higher plasma antibody titers to HIV-1 Env, and by RNA-seq analysis, shared 4 upregulated transcripts compared with control mAb treated monkeys (Fig. 1c & Supplementary Fig. 3C). Two transcripts that were significantly upregulated in both CTLA-4 and CTLA-4 + PD-1 animals B cells compared to controls, Delta-like-1 (*DLL1*) and Colony Stimulating Factor 1 (*CSF1*), have been demonstrated to enhance B cell survival and activation but also may play a role in activating other immune cell populations[41,42]. *ZNF532* and *FAM89A* are predicted transcriptional regulators but have not been functionally characterized in B cells. These data suggested that B cell signaling through Notch and CSF1, in addition to improved CD4 T cell responses, may promote B cell activation and survival needed for higher HIV-1 Env antibody titers in CTLA-4-treated monkeys.

**Immune-checkpoint blockade increased activated T cells.** Lymph node (LN) biopsy samples were collected from all monkeys from both *Macaque studies no. 1 and no. 2* to determine frequencies of germinal center B cells, Tfh and Tfr cells using flow cytometry (Fig. 4a). Seven days after the second CH505 gp120 Env immunization and CTLA-4 or PD-1 Ab antibody co-administration, LN samples were collected from all monkeys (*Macaque study no. 1*). There were no significant differences in the proportion of germinal center (GC) B cells in the LN between CTLA-4, PD-1, CTLA-4 + PD-1 and control Ab treated animals with frequencies of GC B cells averaging between 1.7 and 2.7% of total B cells in each group (Fig. 4b). There was also no statistically significant difference between the proportions of Tfh cells between the antibody treatment groups, with group average frequencies of Tfh cells between 1.5-2.3% of CD4+ T cells (Fig. 4c). Animals treated with PD-1 Ab had higher average frequencies (NS, Wilcoxon–Mann–Whitney) and animals treated with CTLA-4 + PD-1 Abs had significantly

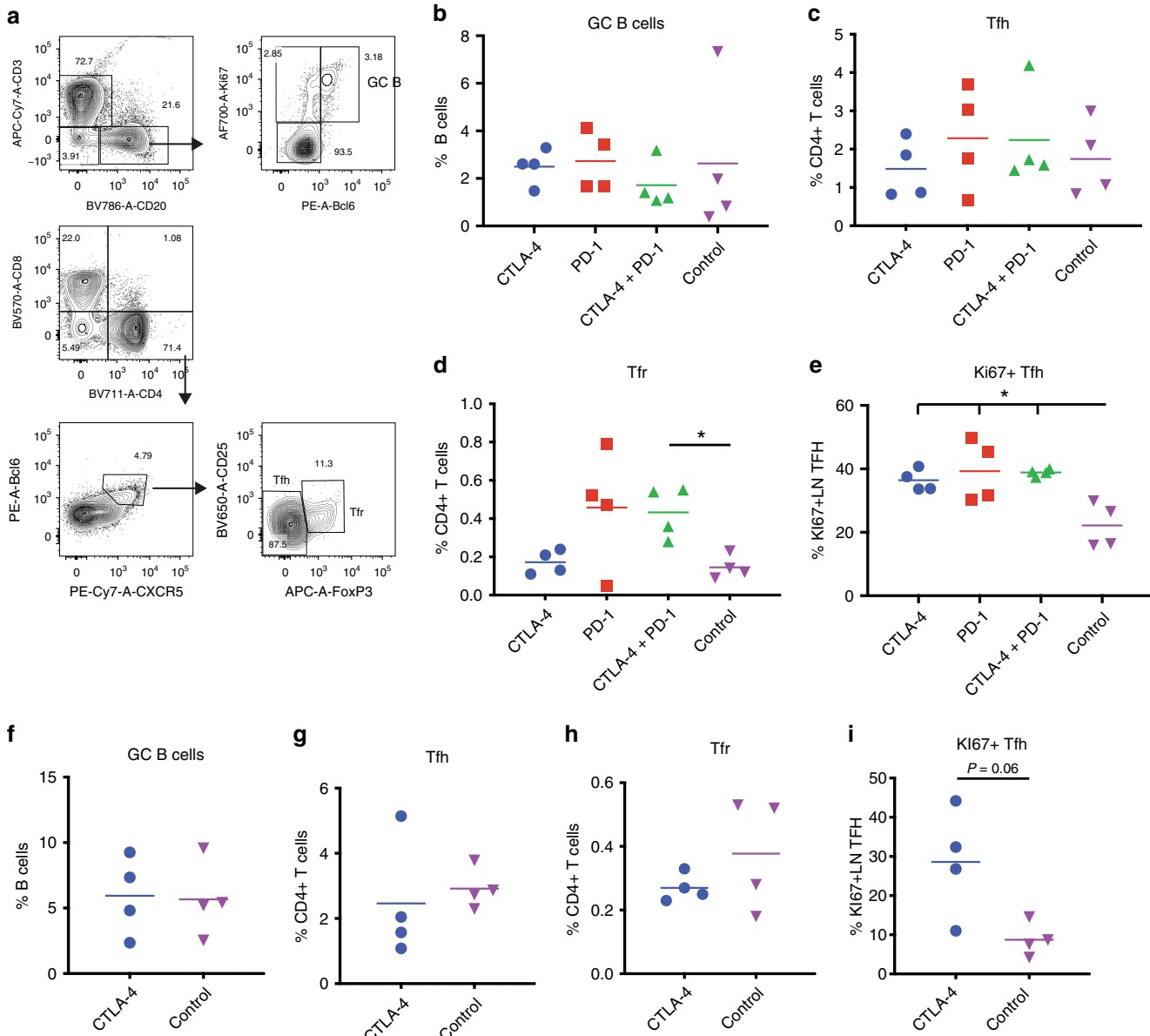

**Fig. 4 Immune checkpoint blockade increases the number of activated T cells in lymph node. a** Representative flow cytometry gating strategy to classify germinal center B cells, Tfh and Tfr cells in lymph node samples. **b–e** Percentage of (**b**) germinal center B cells (**c**) CD4+ T follicular helper cells (**d**) CD4+ T follicular regulatory cells in the axillary lymph node 1 week after the 2nd immunization and co-administration of checkpoint blockade antibody (week 5; NHP study #1). **e** Percentage of Ki67+ CD4 Tfh cells in each group. Points represent individual animals and color designates antibody treatment group (*P < 0.05; Wilcoxon–Mann–Whitney; NHP study#1). **f–i** Percentage of (**f**) germinal center B cells (**g**) CD4+ T follicular helper cells (**h**) CD4+ T follicular regulatory cells in the axillary lymph node 1 week after the 3rd immunization and 2nd co-administration of checkpoint blockade antibody (week 11; NHP study #2). **i** Percentage of Ki67+ CD4 Tfh cells in each group. Points represent individual animals and color designates antibody treatment group (*P < 0.05; Wilcoxon–Mann–Whitney; NHP study#2). Source data are provided as a Source Data file.

(P = 0.03, Wilcoxon–Mann–Whitney) higher frequencies of Tfr cells compared with control Ab treated animals with greater than 2.8-fold average frequency of Tfr in PD-1 or combination treated groups (Fig. 4d). These data suggested that PD-1 blockade increased the proportions of T regulatory cells in draining lymph nodes. Ki67 is expressed on activated and proliferating T cells. Animals treated with CTLA-4, PD-1 and CTLA-4 + PD-1 all had significantly higher proportions of Ki67 + Tfh cells (*P < 0.05, Wilcoxon–Mann–Whitney; Fig. 4e).

Similarly, LN biopsies were obtained seven days after three immunizations and two antibody co-administrations from all animals in each group in the CH848 SOSIP trimer immunized

animals (Macaque study no. 2). Similar to Macaque study no. 1, there were no significant differences in the frequencies of GC B cells, Tfh cells or Tfr cells between CTLA-4 Ab-treated and control Ab treated macaques (Fig. 4f–h). Animals treated with CTLA-4 Ab had a trend of higher average frequencies of Ki67+ CD4 Tfh cells with CTLA-4 Ab-treated macaques just missing the significance threshold (P = 0.06, Wilcoxon–Mann–Whitney, Fig. 4i). Thus, CTLA-4 Ab treatment did not significantly modify the proportions of GC B, Tfh or Tfr cells, but there were alternations in Tfr cell frequencies in the PD-1 Ab treatment group. Treatment with all immune checkpoint Abs resulted in higher average proportions of Ki67+ T cells after immunization.

**Measurement of overall health biomarkers during blockade.**
Immune checkpoint blockade using therapeutic antibodies
against CTLA-4 and PD-1 for cancer immunotherapy results in
autoimmune clinical syndromes in a proportion of indivi-
duals[21,43]. None of the macaques treated with CTLA-4 Ab had
manifestations of clinical autoimmune disease or any adverse
events related to drug or immunization administration during the
study. One macaque in the PD-1 Ab treatment group had to be
euthanized during the study after all immunizations were com-
pleted and histopathology revealed evidence of inflammatory
ulcerative dermatitis in a tail lesion. Moreover, we performed
blood chemistry and hematology tests throughout each macaque
study, and most tests were within normal range with the excep-
tions noted below (Supplementary Figs. 4 and 5). After vaccina-
tion at week 6, there were trends for lower red blood cell,
hemoglobin and hematocrit levels in all macaques with no sig-
nificant differences detected between control Ab and CTLA-4 or
PD-1 treated macaques indicating this trend was unrelated to Ab
treatment and was likely due to frequent blood and tissue sam-
pling during study (Supplementary Fig. 4). Animals treated with
PD-1 Ab had a modest reduction in white blood cell counts after
vaccination that was significantly lower than control Ab treated
animals ($P < 0.03$, Wilcoxon–Mann–Whitney; Supplementary
Fig. 4).

Finally, to determine if evidence could be obtained suggesting
immune tolerance controls were altered by CTLA-4 or PD-1 Ab
treatment, we analyzed plasma antibody titers from the
immunized cynomolgus monkeys from *Macaque study no. 1*
against human autoantigens (autoAg) (dsDNA, Cent-B, Histone,
Jo-1, SSA, SSB, Scl-70, Sm, and RNP) (Supplementary Fig. 1H).
No animals had detectable pre-existing positive autoantibody
titers to these antigens and control animals remained negative
throughout immunization (Supplementary Fig. 1H). For monkeys
treated with CTLA-4 or PD-1 Ab after one immunization, two
CTLA-4 + PD-1 Ab-treated macaques were positive for Abs to a
single autoAg (SCL-70 and RNP antigens, respectively), and after
2 immunizations, one PD-1 and one CTLA-4 + PD-1 Ab-treated
monkey exhibited reactivity to a single autoAg (SCL-70 antigen).
After 4 immunizations, 5 of the 8 animals treated with CTLA-4 or
CTLA-4 + PD-1 Ab, respectively, had positive autoAg antibody
titers to Scl-70, histone and RNP antigens (Supplementary
Fig. 1H). Thus, transient administration of CTLA-4 or PD-1
Ab treatment was associated with macaque plasma autoantibody
seropositivity.

**Modulation of CTLA-4 or OX40 increased VRC01 bnAb
responses.** To study VRC01-class, CD4 binding site bnAb
development[44], a mouse model was engineered to produce
diverse repertoires of VRC01-like precursors. In this model,
human VH1-2*02 segment can recombine with the whole set of
mouse D segments and the human $J_H2$ segment to create a
repertoire of diverse Ig heavy chains with a wide range of CDR
H3s (Supplementary Fig. 6A). This model is similar to a pre-
viously reported version, but with the additional replacement of
the mouse $J_H$ segments with the human $J_H2$ segment[45]. Human
$J_H2$ segment can contribute a conserved tyrosine residue in the
CDR H3 of VRC01-class antibodies, and the incorporation of
human $J_H2$ segment may increase the frequencies of VRC01-like
precursors in this mouse model. The VH1-2 heavy chains pair
with a rearranged germline form of the VRC01 kappa light chain
(gl-VRC01-LC), which contains the signature five-amino acid
CDRL3 of VRC01-class antibodies necessary for VRC01-like
bnAb precursors (Supplementary Fig. 6A). We utilized a stepwise
immunization regimen to engage the VRC01-like germline pre-
cursors and to select for neutralizing capability, as previously

described[45]. Specifically, we primed with eOD-GT8-60mer par-
ticles that have been demonstrated to engage VRC01-class pre-
cursors[46], then boosted five-times with a series of immunogens
starting with a 426c gp120 core Env with glycans deleted near the
CD4 binding site (N276, N460, N463; degly3) multimerized on a
ferritin particle; subsequent immunogens were the 426c core Env
protein with serial additions of deleted glycans (426c-degly2 and
426c-degly1, the latter lacking N276 only) followed by the fully
glycosylated 426c gp120 wt core and finally, the 426c-WT native-
like SOSIP trimer[45,47]. Along with the last two immunizations of
the 426c gp120 wt core and 426c-WT SOSIP trimer, we admi-
nistered CTLA-4 blocking antibody ($n = 4$, Group 1), OX40
agonist antibody ($n = 4$, Group 2), or a control group that
received isotype control antibody ($n = 2$); or saline alone ($n = 2$).
These animals were combined as the control group. All animals
were harvested a week after the last immunization with the 426c-
WT native-like SOSIP trimer to measure blood and tissue
immune responses (Fig. 5a).

After the 426c-WT native-like SOSIP trimer immunization
(at week 11), the highest average serum IgG ELISA titers were
observed for the eOD-GT8 priming antigen, but measurable
IgG titers were detected against all vaccine antigens in all mice
for all groups (Fig. 5b). CTLA-4 Ab and OX40 Ab groups had
higher average serum antibody titers against all vaccine
immunogens when compared to controls, and the CTLA-4
Ab-treated group had statistically significantly higher titers of
antibodies against 426c-degly3 and 426c-degly1 compared to
controls ($P = 0.03$; Wilcoxon–Mann–Whitney; Fig. 5b). More-
over, we found significantly higher serum IgG titers against the
native-like 426c SOSIP trimer in mice given CTLA-4 Ab ($P =
0.03$; Wilcoxon–Mann–Whitney) or OX40 Ab ($P = 0.03$;
Wilcoxon–Mann–Whitney); significantly higher IgG responses
to the heterologous BG505 SOSIP were also observed in the
CTLA-4 Ab group ($P = 0.03$; Wilcoxon–Mann–Whitney;
Fig. 5c). In contrast, mice in the control group had little or
no detectable serum IgG for autologous, 426c-WT, or hetero-
logous, BG505, SOSIP trimers (Fig. 5c). These results indicated
better induction of IgG bnAb precursors that recognized native-
like HIV-1 Env trimers in those groups receiving CTLA-4 or
OX40 Abs although the titers against the SOSIP trimers were
several logs lower than the priming immunogens. CTLA-4 Ab
and OX40 Ab-treated animals had greater than 3-fold higher
average neutralizing antibody titers against the 426c virus 3
glycan mutant (426c_Degly3) and the 426c virus lacking only
the N276 glycan (426_Degly1) with OX40-Ab treated animals
having significantly higher titers compared with controls ($P =
0.03$; Wilcoxon–Mann–Whitney; Fig. 5d, e).

Next, we quantitated the frequencies of B and T cell
populations in the inguinal lymph nodes (LN) of immunized
mice that are critical for the bnAb response at week 11 after the
final immunization using flow cytometry. GC and memory B
cell frequencies were determined for each animal (Supplemen-
tary Fig. 6B). CTLA-4 antibody-treated animals had signifi-
cantly higher GC B cell frequencies when compared to control-
immunized or naïve (unimmunized) animals ($P = 0.03$;
Wilcoxon–Mann–Whitney; Fig. 6a). Animals treated with
OX40 Ab also had higher average frequencies of GC B
cells but missed the significance threshold ($P = 0.06$;
Wilcoxon–Mann–Whitney). CTLA-4 Ab and OX40 Ab treated
groups also had significantly higher frequencies of memory B
cells ($P = 0.03$; Wilcoxon–Mann–Whitney; Fig. 6b).

Germinal center Tfh cells stimulate GC B cells whereas Tfr cells
dampen the B cell response and can be distinguished by
FoxP3 surface marker expression that is expressed on Tfr cells
(Supplementary Fig. 6C). Both CTLA-4 Ab or OX40 Ab-treated
animals had significantly higher frequencies of GC Tfh cells and

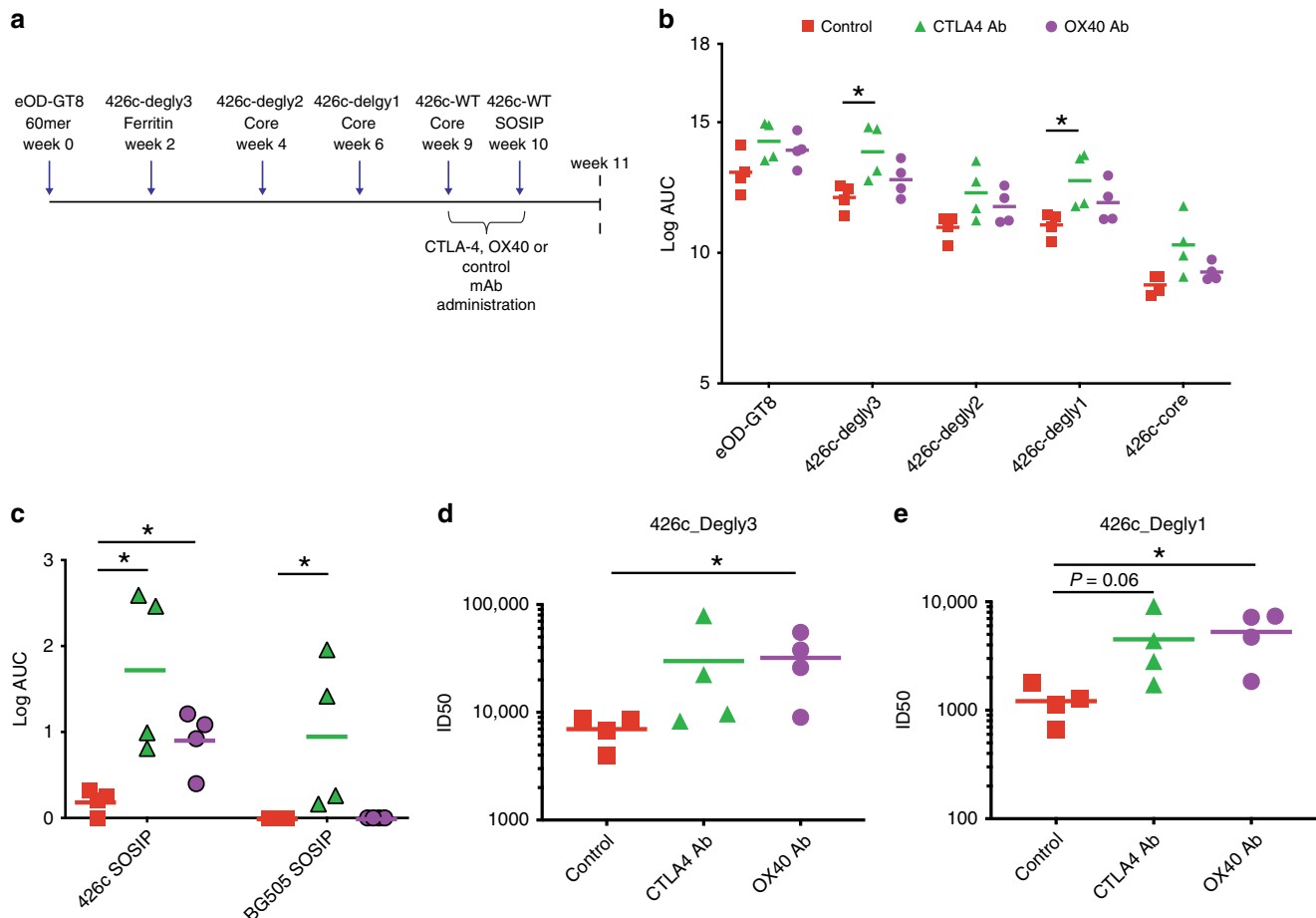

**Fig. 5 Co-administration of CTLA-4 or OX40 with precursor targeting vaccine in VRC01 knock-in mice. a** Vaccination protocol for immunization of four groups of VRC01 precursor knock-in mice with sequential germline-targeting vaccine regimen with co-administration of CTLA-4 blocking (CTLA4 Ab; $n =$ 4) or OX40 agonist (OX40 Ab; $n = 4$) or control ($n = 4$) antibodies. Additionally, we included a no co-administration group ($n = 2$) as control. **b, c** Plasma antibody titers at week 11 of (**b**) binding to sequential vaccine proteins by ELISA measured by Log area under the curve (**c**) binding to autologous (426c) and heterologous (BG505) SOSIP trimers by ELISA measured by Log area under the curve. **d, e** Plasma antibody neutralization of (**d**) 3 glycan deleted 426c virus (426c_Degly3) and (**e**) N276 glycan deleted (426c_Degly1) at week 11 measured in the TZM-bl assay. Inhibitory dilution at 50% (ID50) graphed. Each dot represents a single animal and are colored based on the antibody treatment group (Control Ab, red; CTLA4 Ab, green; OX40 Ab, purple). *$P \leq$ 0.05, Wilcoxon–Mann–Whitney. Source data are provided as a Source Data file.

lower frequencies of Tfr cells compared to control ($P = 0.03$; Wilcoxon–Mann–Whitney; Fig. 6c, d). Thus, CTLA-4 blockade as well as OX40 agonism augmented VRC01-bnAb lineage antibody responses in a humanized mouse model, and did so by increasing GC B cell and Tfh cells while reducing Tfr cell proportions in the lymph node.

We isolated GC B cells by sorting CD38lowGL7+ B cells from the inguinal LN samples (Supplementary Fig. 6A) and performed high-throughput heavy chain and light chain immunoglobulin (Ig) repertoire sequencing to determine the antibody repertoire[48]. We found the control group (no treatment and control antibody treatment) and OX40 Ab group had a higher frequency of utilization of the IGHV1-2*2 heavy chain gene sequences (mean 28.4% and 21.8%, respectively) compared to the CTLA-4 Ab treated group (mean 15.7%; $P = 0.03$, Wilcoxon–Mann–Whitney; Fig. 6e and Supplementary Fig. 6D). In animals in all 4 groups, greater than 68% of the IGHV1-2*02 DNA reads from the GC were mutated (Supplementary Fig. 6D). Among the mutated IGHV1-2*02 sequences, CTLA-4 Ab-treated animals had significantly lower, whereas OX40-Ab treated animals had significantly higher, average mutation frequencies compared to control animals ($P < 0.001$, Wilcoxon–Mann–Whitney), although the range of minimum and maximum mutation frequencies were

similar between the groups (Fig. 6f and Supplementary Fig. 6D). Functional heavy (N54R/T, S55R, T58VP) and light chain (CDR1 SY 2AA deletion) mutations that occurred in mature VRC01-class antibodies were identified in all groups.

Since only a fraction of the DNA reads (12.9–36.6%) of the GC B cells were derived from the human knock-in IGHV1-2*02 gene, and the CTLA-4 Ab treatment decreased the observed frequency of this gene (Fig. 6e and Supplementary Fig. 6D), we analyzed the genetics of the non-VH1-2*02 i.e. the mouse IgH repertoire of the GC B cells. We found that there were no significant differences in the distribution of CDR H3 lengths or median mutation frequencies of the mouse *Ig* genes between the antibody treatment or control groups (Supplementary Fig. 6D). Next, we determined the frequency of mouse variable heavy gene segment usage in the GC B cells from each group, and we found that the control and OX40 Ab-treated group used a diverse array of IGHV segments with IGHV3-1 and IGHV2-6 being among the most frequent (Supplementary Fig. S6E). Surprisingly, 15.5% and 18.7% of the CTLA-4 Ab treated groups sequences utilized IGHV2-6 and IGHV3-1, indicating a more selected mouse Ig repertoire (Supplementary Fig. 6E). This is reflected in lower entropy value that is a measure of overall gene usage diversity of 3.26 in the CTLA-4 Ab treated animals compared to control groups (3.66

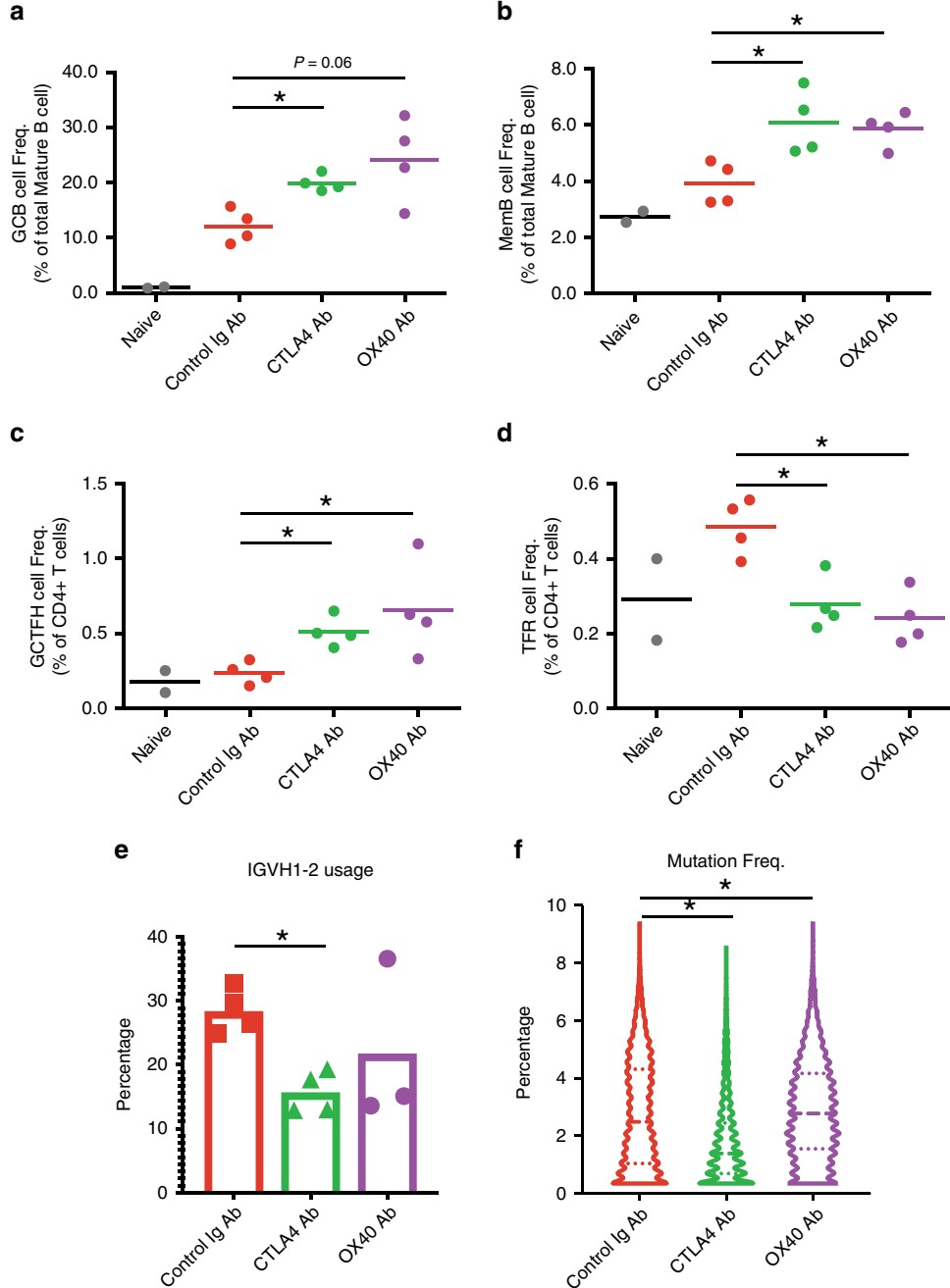

**Fig. 6 Co-administration of CTLA-4 or OX40 altered splenic germinal center cell frequencies and repertoire in VRC01 precursor mouse after immunization. a–d** Frequencies of splenic (**a**) Germinal center B cells and (**b**) memory B cells (**c**) germinal center CD4+ T follicular helper cells and (**d**) germinal center T follicular regulatory cells in each group after immunization at week 11. Each dot represents a single animal and are colored based on antibody treatment group (Control Ab, red; CTLA4 Ab, green; OX40 Ab, purple). **e, f** Sequencing of germinal center B cell IGHV genes by HTGTS-Rep-Seq. **e** Fraction of GC B cells HTGTS-Rep-Seq reads that utilize the human IGHV1-2*02 gene. Each dot represents the individual animals and the bars indicate the group means. **f** Violin plots of the frequency of mutation from germline of IGHV1-2*02 reads that contain at least a single mutation. *$P < 0.05$, Wilcoxon–Mann–Whitney. Source data are provided as a Source Data file.

and 3.65; Supplementary Fig. 6E). Thus, treatment of VRC01 precursor KI mice with CTLA-4 Ab resulted in an altered GC B cell repertoire with increased usuage of select endogenous mouse heavy chain genes, whereas OX40 Ab treatment augmented somatic hypermutation frequency of the human IGVH1-2 bnAb precursor gene.

**Identification of LN transcriptome networks by scRNA-seq.** We next performed single-cell RNA-sequencing (scRNA-seq) on

a total of 25,843 lymph node cells from 2 VRC01 KI mice each from control, CTLA-4 Ab and OX40 Ab-treated animals after final immunization with the 426c-WT native-like SOSIP trimer. Dimensionality reduction by tSNE and graph-based clustering identified 13 distinct transcriptional clusters (Fig. 7a). By using known transcripts and determining significantly upregulated genes in each cluster, we were able to identify major populations of immune cells such as B cells, CD4+ and CD8+ T cells as well as less common immune cell populations, such as monocytes,

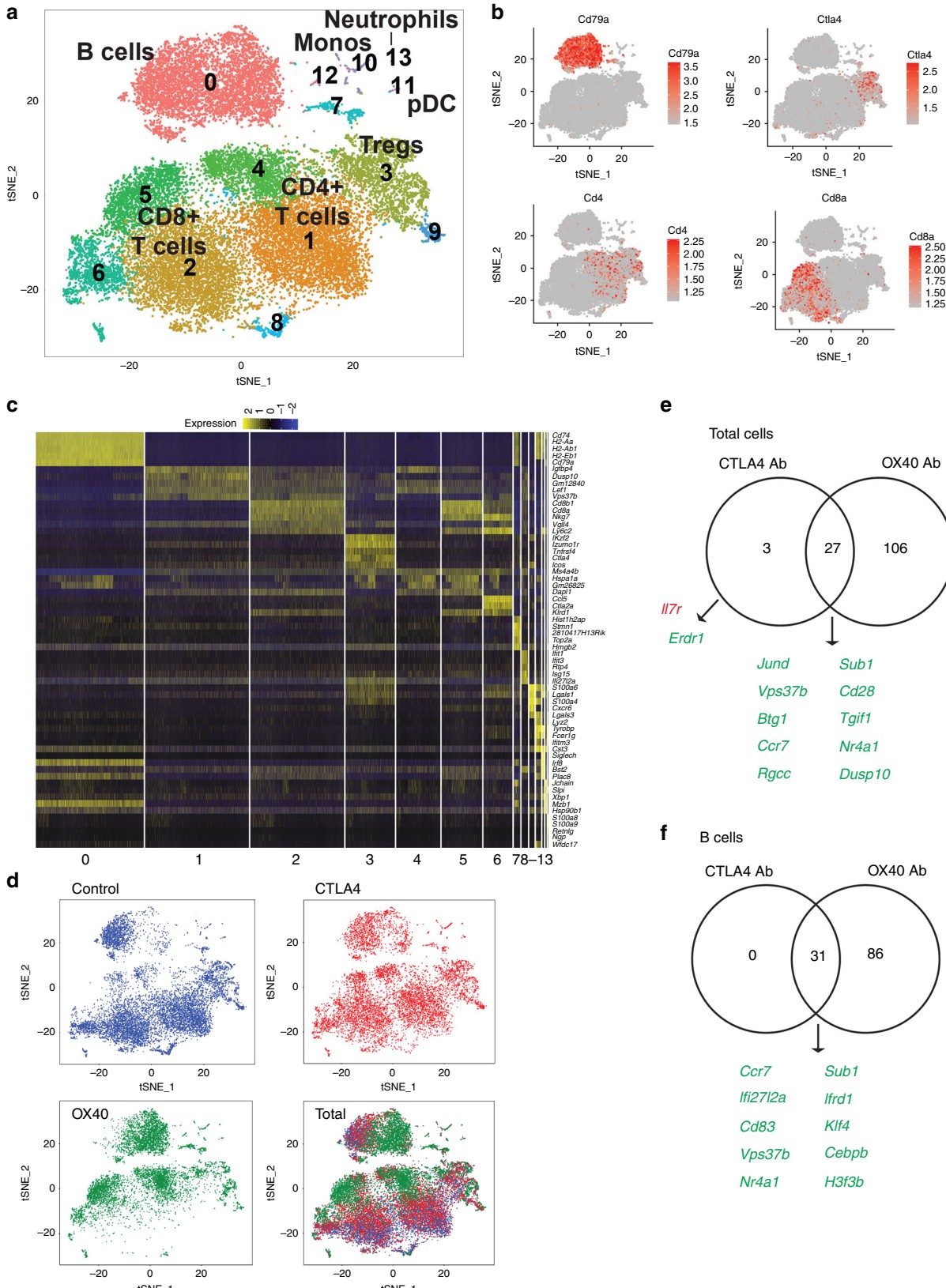

dendritic cells, and neutrophils (Fig. 7b, c). Cells within immune cell clusters from the CTLA-4 Ab (red) and control antibody-treated (blue) mice clustered near each other, indicating overall similarity of the individual cell transcriptional profiles (Fig. 7d). In contrast, cells from the OX40 Ab-treated mice (green) formed

distinct clusters in the B cell, CD4 and CD8 T cell immune cell clusters compared to control cells (blue), suggesting distinct transcriptional changes in OX40 Ab-administered mouse cells (Fig. 7d). Significantly changed transcripts were determined by comparing CTLA-4 and OX40 Ab to control antibody cells and

**Fig. 7 scRNA-seq of lymph node cells from immunized VRC01 knock-in mice. a** Graph-based clustering and t-distributed stochastic neighbor embedding (tSNE) dimensionality reduction plots of single-cell RNA-seq of 10,226 cells from 2 control, 7679 cells from 2 CTLA-4 Ab treated and 7938 cells from 2 OX-40 Ab treated mice lymph node tissue after the completion of immunization. 13 distinct transcriptional clusters were identified on the aggregated cells. **b** Feature plots of the expression of individual transcripts on the tSNE plot to identify B cells (*Cd79a*), CD4 T cells (*Cd4*), CD8 T cells (*Cd8a*) and activated/ regulatory T cells (*Ctla4*). **c** Heatmap of the top 5 upregulated genes in each of the 13 transcriptionally identified clusters. Expression of individual genes shown for 100 cells for each cluster or all cells if less than 100 cells in cluster. **d** tSNE plots showing cells from control mice (blue), CTLA4 Ab treated mice (red), OX40 Ab treated mice (green) and all groups together (total). **e, f** Venn diagram of significantly changed genes determined using a likelihood-ratio test (*P* < 0.05) comparing all cells (**e**) or only B cells (**f**). Selected transcripts identified that are upregulated compared to control (red font) or downregulated (green).

consisted of 30 and 133 differentially expressed transcripts in the CTLA-4 and OX40 Ab groups, respectively (Fig. 7e; Supplementary Data 7 & 8). Twenty-seven of the 30 transcripts changed in the CTLA-4 Ab group were also changed in the OX40 Ab group; the CTLA-4 Ab group had the unique upregulation of *Il7r* that encodes the IL7 receptor and downregulation of *Erdr1* that regulates cellular apoptosis[49]. Among the 27 differentially-expressed transcripts shared between CTLA-4 Ab and OX40 Ab treatments, there was downregulation of transcription factors (*Jund, Sub1, Tgif1*) that control cell death and proliferation[50–52], including downregulation of *Nr4a1* which has been shown to promote regulatory T cell differentiation[53]. *Btg1, Dusp10,* and *Rgcc* also are factors that regulate apoptosis (Fig. 4e)[54–56]. Downregulation of *Ccr7* is important for CD4 T cell migration to the T-B border in lymphoid tissues[57], and *Cd28* and *Vps37b* are important factors for T cell activation involved in CTLA-4 and OX40 pathways[58,59].

Next, we analyzed genes in B cells (cluster 0, *Cd79a* positive) that were differentially expressed between CTLA-4 or OX40 Ab-treated mice compared with control animals (Fig. 7f). Thirty-one (Supplementary Data 9) and 117 (Supplementary Data 10) significantly changed genes were identified in CTLA-4 Ab and OX40 Ab-treated mice, respectively, with all of the B cell gene transcripts changed in CTLA-4 Ab-treated mice also downregulated in OX40 Ab treated mice (Fig. 7f). Genes downregulated in CTLA-4 Ab-treated B cells included not only transcripts observed when comparing all of the cells (Fig. 7e), but also there were additional transcripts down-regulated that are important regulators of B cell activation such as *Cd83* and *Klf4*[60,61]. These data indicated that CTLA-4 Ab and to a greater extent OX40 Ab-treated cells downregulated pro-apoptotic factors and upregulated transcripts important for cellular activation and proliferation that may have contributed to the increased germinal center Tfh and B cell frequencies and antibody responses in bnAb KI mice.

## Discussion
A critical goal of this study was to determine if Env antibody responses to HIV-1 envelope could be enhanced with immune checkpoint inhibitors or agonists for activating receptors. Both CTLA-4 blockade and an OX40 agonist increased HIV-1 Env-induced neutralizing antibody B cell responses in bnAb knock-in mice, with similar trends of CTLA-4 blockade in outbred cynomolgus monkeys. Importantly, both CTLA-4 and OX40 Ab treatment expanded GC B and Tfh cells in the VRC01 CD4 binding site bnAb precursor mouse model indicating improvement of the germinal center responses in this model with engineered B cell specificity and increased frequency. Changes in the proportions of GC B cells or Tfh cells were not observed in the macaque model and may be due to the polyspecific antibody response and/or timing and location of lymph node sampling. Future studies using longitudinal sampling and antigen-specific cell enumeration will be required to determine the impact of

immune checkpoint modulation of antigen-specific cell frequencies in primates.

Expression of co-inhibitory receptors such as CTLA-4 is a hallmark of T regulatory cells[62]. CTLA-4 Ab-mediated germinal center enhancement likely occurs through inhibition of interactions between CTLA-4 on T regulatory cells CD80 and CD86 on antigen presenting cells[19]. In contrast, the mechanism of OX40 antibody enhancement of Tfh and GC B cell responses is stimulation of OX40 on CD4 Tfh[63]. OX40 has been demonstrated to amplify Tfh development and GC reactions during vaccinia infection[33]. Moreover, OX40 has been shown to inhibit the development and function of T regulatory cells[64,65]. Single-cell RNA-seq of lymph node cells showed that mice treated with CTLA-4 Ab, and to a greater extent those treated with OX40 Ab, had significant changes to transcriptome networks in their immune cell populations. OX40 Ab treated animals had immune cells with increased markers of activation, proliferation and germinal center regulation. This analysis revealed the molecular pathways that are activated by the CTLA-4 and OX40 Ab treatments in the LN during immunization and identify other molecules that may be targeted for manipulation to improve GC responses during vaccination.

Notably, PD-1 antibody treatment in macaques did not enhance HIV-1 Env antibody responses. Rather, in the PD-1 Ab treated animals, we observed increased proportions of Tfr cells in the lymph nodes, suggesting that PD-1 blockade enhanced regulatory mechanisms that inhibited Env antibody responses after immunization[32]. Indeed, transcriptome analysis of CD4 T cells identified increased expression levels of transcripts associated with T cell activation and proliferation in the CTLA-4 Ab-treated monkeys and increased expression levels of T cell inhibitory transcripts in the PD-1 Ab-treated monkeys. PD-1 signaling is also critical for Tfh signaling, so early blockade of this signaling axis could have inhibited vaccine-induced antibody generation. Thus, future studies investigating the timing of immune checkpoint blockade on antibody responses are required.

BnAb generation has been suggested to be controlled in part by immune tolerance mechanisms due to polyreactivity, autoreactivity and/or long heavy chain complementarity determining regions[13,14,66]. In macaques, treatment with CTLA-4 or a combination of CTLA-4 + PD-1 antibodies resulted in induction of plasma autoantibodies to SCL-70, RNP and Histone proteins commonly observed in human autoimmunity. Thus, these treatments, as in humans, broke tolerance to autoantibody production, and also expanded bnAb GC expansion in bnAb KI mice, but did not lead to full bnAb affinity maturation.

BnAbs develop after years in ~50% of individuals who are HIV-1 infected[15], but have not been induced in humans in the setting of vaccination[67]. Study of those HIV-1-infected individuals who make bnAbs vs. those who do not, have demonstrated that those who make bnAbs have a higher frequency of serum autoantibodies, higher frequencies of circulating memory Tfh[16,17], and lower frequencies of T regulatory cells, with those Tregs that are present being PD-1 positive with an exhausted

phenotype[17]. Moreover, when bnAbs develop with maximum antigenic stimulation during infection, it takes 3-5 years for full bnAb generation with high levels of somatic hypermutation in the antibody genes. CTLA-4 and OX40 Abs had only modest effects on SHM frequency, thus immunotherapies specifically targeting B cell SHM may be needed. In addition, bnAbs have been reported to develop more readily in the setting of autoimmune disease such as systemic lupus erythematosus, but even in that setting, bnAbs do not develop immediately, but only after years of stimulation[68]. While tolerance was interrupted in the setting of CTLA-4 and PD-1 antibody administration to macaques, bnAbs were not induced, indicating the need for specifically designed sequential Envs to overcome additional limitations on induction of bnAbs, including the development of Envs to select for improbable mutations that are common in bnAb B cell lineages[69,70].

Safety of any vaccine administered is of paramount importance. While any clinical autoimmune side effects of such treatments will be unacceptable in the setting of vaccination of healthy individuals, it is possible that transient treatment with these or similar immunomodulators may allow for expansion of bnAb precursors by UCA-targeting Envs, thus allowing sequential Env immunizations to be more successful in selecting for affinity mature bnAbs. When CTLA-4 Ab is frequently administered over long periods of time when treating malignancy. Here, we show that CTLA-4 Ab treatment can enhance HIV-1 neutralizing antibody induction after only two or three administrations. We found that in the bnAb KI mouse CTLA-4 Ab treatment did alter the GC B cell repertoire and CTLA-4 and/or PD-1 treated macaques were associated with increase in serum autoAgs. In contrast, OX40 treated mice did have an increase in GC B cells, they did not exhibit an increase in GC VH sequences that were not of bnAb origin. Moreover, OX40 Ab-treated mice had an increased fraction of GC VH sequences that had increased mutation frequency compared to control groups. While OX40 Ab treatment can facilitate tumor rejection in mice[71], blockade of OX40 has been associated with a reduction in CD4-driven autoimmunity[72]. Thus, any use of CTLA-4 or OX40 Abs as adjuvants for Env vaccination will have to be used transiently and it will be important to clearly demonstrate that, as used, there is no induction of manifestations of autoimmune disease. Design of delivery systems to deliver or express CTLA-4 antibody or OX40 Abs locally in draining LNs may serve to localize and limit systemic effects of checkpoint inhibition.

Overall, several vaccine traits will need to be optimized simultaneously to induce full development of broadly neutralizing antibodies. Optimizations required include: comparing Env structure (including multimerization and/or presentation on viral membranes), sequential Env designs to "guide" desired but usually disfavored and rare bnAb B cell lineages with improbable mutations, comparing modes of immunization (protein, DNA, and mRNA), and altering adjuvant formulations[67,69,70,73]. Here, we have shown that one additional factor that is plausible to consider is the addition of immune modulators to enhance titers of neutralizing antibody responses in HIV-1 Env vaccination.

## Methods

**Cynomolgus macaque immunization.** Recombinant gp120 proteins CH505 TF, CH505wk53, CH505wk78, and CH505wk100 were produced by KBI Biopharma by expression in CHO cells (KBI Biopharma) and capture by lectin affinity columns[74,75]. We immunized 16 Cynomolgus macaques in 4 groups sequentially with the CH505 TF and natural variants (wk53, wk78, and wk100) every 4 weeks with 100 μg of protein administered intramuscularly in GLA-SE (IDRI-EM107) adjuvant. Selected animals were randomly assigned to each vaccine arm. Group 1 animals ($n = 4$) were coadministered 10 mg/kg Ipilimumab (Bristol-Myers-Squib) during the immunization 1–3. Group 2 animals ($n = 4$) were coadministered

10 mg/kg Nivolumab (Bristol-Myers-Squib) during the immunization 1–2. Group 3 animals ($n = 4$) were coadministered 10 mg/kg Ipilimumab and 10 mg/kg Nivolumab (Bristol-Myers-Squib) during the immunization 1–3. Group 4 animals ($n = 4$) were coadministered 10 mg/kg anti-influenza (CH65; Duke) during the immunization 1–3. For the second Cynomolgus macaque study, we made recombinant SOSIP envelopes by transient transfection. Freestyle 293 (Life Technologies) cells were cultured in Freestyle 293 media below $3 \times 10^6$ cells/mL. On the day of transfection, cells were diluted to $1.25 \times 10^6$ cells/mL with fresh media and 1 L of cells was transfected with 293Fectin (Life Technologies) complexed with 650 μg envelope-expressing DNA and 150 μg of furin expressing plasmid DNA. Cells were cultured for 6 days in shaker flasks. Cell culture supernatant was cleared of cells by centrifugation for 30 min at 3500 rpm and subsequently 0.8 μm filtered. The cell-free supernatant was concentrated to less than 100 mL with a single-use tangential flow filtration cassette and 0.8 μm filtered again. Trimeric Env protein was purified with PGT145 affinity chromatography. One hundred mg of PGT145 IgG1 antibody was conjugated to 10 mL of CnBr-activated sepharose FastFlow resin (GE Healthcare). Coupled resin was packed into Tricorn column (GE Healthcare), and stored in PBS supplemented with 0.05% sodium azide. Cell-free supernatant was applied to the column at 2 mL/min in PBS supplemented with 0.05% sodium azide using an AKTA Pure (GE Healthcare). The column was washed, and protein was eluted off of the column with 3 M MgCl₂. The eluate was immediately diluted in 10 mM Tris pH8, 0.2 μm filtered, and concentrated down to 2 mL for size exclusion chromatography. To produce biotinylated CH0848 10.17DT SOSIP gp140s, the envelope sequence was expressed with a C-terminal avidin tag (AviTag: GLNDIFEAQKIEWHE). After antibody affinity chromatography and eluate concentration, the envelope was dialyzed for 1 h in 10 mM Tris pH8. Envelope was biotinylated with the BirA biotin-protein ligase standard reaction kit (Avidity). The ligation reaction was done by agitating 25 μM of SOSIP trimer at 900 rpm at 30 °C for 5 h. The biotinylated protein was then concentrated to 2 mL for size exclusion chromatography. Size exclusion chromatography was performed with a Superose6 16/600 column (GE Healthcare) in 10 mM Tris pH8, 500 mM NaCl. Fractions containing trimeric HIV-1 Env protein were pooled together, sterile-filtered, snap frozen, and stored at −80 °C. Eight Cynomolgus macaques were immunized with sequential SOSIP trimers and coadministered antibodies in 2 groups ($n = 4$/group) with 10 mg/kg Ipilipumamb (Bristol-Myers-Squibb), or control CH65 anti-HA at 10 mg/kg by IV infusion. Monkeys were housed at Bioqual, Rockville, MD. The animals were maintained in accordance with the National Institutes of Health and Duke University guidelines and all studies were approved by the appropriate Institutional Animal Care and Use Committee.

**Antibody binding and neutralization.** Binding of vaccine plasma antibodies to HIV-1 Envs or autoantigens was measured by standard enzyme-linked immunosorbent assays (ELISA) and binding titers reported as log area under the curve (AUC)[38,76,77]. Neutralization activities of monkey and mouse plasma and purified antibodies were determined by the TZM-bl-cell-based neutralization assay[78,79]. Neutralization assays are performed in technical triplicate for all animals at each serum time point or antibody concentration.

**Surface plasmon resonance (SPR) measurements of purified plasma IgG avidity.** IgG avidity of purified total IgG samples from sera of immunized animals to a panel of HIV-1 antigens (CH505TFgp120, CH505TFchim.6R.SOSIP.664.avi, CH848.3.D0949.10.17chim.6R.DS.SOSIP.664_avi, CH848.3.D0949.10.17chim.6R.DS.SOSIP.664_N133D_N138T_avi) was measured by surface plasmon resonance (BIAcoreTM 3000, BIAcore/GE Healthcare, Pittsburgh, PA) analysis as in previous studies[80,81]. Binding responses were measured by SPR following the capture of biotinylated Env SOSIP proteins to immobilized streptavidin on CM5 sensor chips (BIAcore/GE Healthcare, Pittsburgh, PA), with the exception of CH505TF gp120 that was immobilized using standard amine coupling chemistry. Purified plasma IgG samples at 200 μg/ml were flowed (2.5 min) over antigen immobilized chip surfaces followed by a dissociation phase (post-injection/buffer wash) of 10 min and a regeneration with Glycine pH2.0. Non-specific binding of a preimmune sample was subtracted from each post-immunization IgG sample binding data. Data analyses were performed with BIA-evaluation 4.1 software (BIAcore/GE Healthcare, Pittsburgh, PA). Binding responses were measured by averaging post-injection response unit (RU) over a 10 s window; and dissociation rate constant, kd (second-1), was measured during the post-injection phase after stabilization of signal. Positive response was defined when both replicates have a RU value ≥10. Relative avidity binding score is calculated as follows: Avidity score (RU.s) = (Binding Response Units /kd)[80,81].

**RNA-seq of CD4 T cells and CD20 B cells.** PBMCs from cynomolgus macaques were stained with fluorescently labeled antibodies for cell surface markers (see below). CD4 T cells were defined as viable singlet CD45+, CD14−, CD3+ and CD4+ cells and CD20 B cells were defined as CD45+, CD14−, CD20+ cells. Cells were sorted into tubes containing RLTplus (Qiagen) lysis buffer. RNA was purified using the RNAeasy Mini kit according to manufacturer protocol (Qiagen). RNA was quantified using the Agilent Tapestation RNA high-sensitivity kit. Reverse transcription and amplification were carried out using the Smartseq ultra-low v4

kit (Clontech). 200 pg of amplified cDNA was prepared for Illumina sequencing using the Nextera HT library preparation kit (Illumina). Libraries were quantified using the quantitative PCR (Kapa Biosystems) and sequenced to a minimum depth of 25 million reads per sample (2 × 75 bp) on the Illumina NextSeq.

After sequencing fastq files were quality filtered and trimmed using TrimGalore and aligned to the rhesus macaque genome (Rm8) using STAR. After reference alignment read counts on each gene were quantified by HTseq and significantly differentially expressed transcripts were determined by DeSeq2 R package and heatmaps also were generated with DeSeq2 in R.

**Monkey lymph node phenotyping.** Cryopreserved cells were thawed and counted, then labeled for surface antigens with the following fluorochrome-antibody conjugates: FITC anti-human CD45RA (clone MEM-56; Thermo Fisher #MA1-19570; 1:20 dilution for staining), PE-Dazzle594 anti-human CXCR3 (clone 1C6/CXCR3; BD Biosciences #562451; 1:20), PE-Cy5 anti-human CD69 (clone FN50; Biolegend #310908; 1:50), PE-Cy7 anti-human CXCR5 (clone MU5UBEE; Thermo Fisher #25-9185-42; 1:50), APC-Cy7 anti-human CD3 (clone SP34-2; BD Biosciences #557757; 1:50), BV421 anti-human PD-1 (clone EH12.2H7; Biolegend #329920; 1:50), BV570 anti-human CD8 (clone RPA-T8; Biolegend #301038; 1:50), BV650 anti-human CD25 (clone BC96; Biolegend #302634; 1:50), BV711 anti-human CD4 (clone OKT4; Biolegend #317440; 1:50), and BV785 anti-human CD20 (clone 2H7; Biolegend #302356; 1:50). Cells were then incubated with LIVE/DEAD$^{TM}$ Fixable Aqua Dead Cell Stain (ThermoFisher #L34957; 1:1000) to allow exclusion of dead cells. Cells were then fixed and permeabilized using FoxP3 Transcription Factor Buffers (ThermoFisher), and then labeled for intracellular antigens with the following fluorochrome-antibody conjugates: PE anti-human Bcl-6 (clone K112-91; BD Biosciences #561522; 1:50), APC anti-human FoxP3 (clone PCH101; Thermo Fisher #17-4776-42; 1:20), and AF700 anti-human Ki-67 (clone B56; BD Biosciences #561277; 1:50). Data were acquired on a BD LSRII flow cytometer and analyzed using FlowJo version 10.

**Protein production for mouse immunization study.** All proteins were produced using Expi293 or 293F cells. One mg of plasmid DNA per 1 liter of cells was diluted in DMEM and mixed with PEI. PEI:DNA mixtures were added to 293F cells (ThermoFisher, catalog #R79007) for 4 h. 293 F cells were subsequently washed and diluted to 1.25 million cells per ml in Freestyle293 media (ThermoFisher). The cells were cultured for 5 days and on the fifth day the cell culture media was cleared of cells by centrifugation. The cell culture media was concentrated with vivaflow 50 and protein isolated by lectin beads (Vistar Labs). The isolated protein was eluded with 0.5 M methyl-a-pyranoside and buffer exchanged into PBS. The immunogens, eOD-GT8_60mer 426c-degly3 (N276D, N460D, N463D) core-Ferritin, 426c-degly2 (N276D, N460D) core, 426c-degly1 (N276D) core, 426c-WT gp120 core, were purified with *Galanthus nivalis* (GNA)-lectin gel (EY Laboratories, Inc.) and followed with a gel filtration chromatography. The BG505.SOSIP trimer and the 426c-WT.SOSIP trimer were purified with the 2G12/SEC method as previously described[45].

**Immunization of VH1-2/LC mouse models.** Mice expressing the VH1-2 rearranging germline and the rearranged gl-VRC01 light chain were engineered as previously described[45], with the additional substitution of mouse J$_H$1-4 with human J$_H$2 segment[45]. For immunization studies, mice were immunized sequentially with 25 μg of protein per animal in 60 μg of Poly I:C adjuvant (Invivogen) at two sites intramuscularly. For antibody coadministration 250 μg of anti-CTLA-4(BioXCell #BE0131), anti-OX-40 (BioXCell #BE0031) or a combination of isotype controls (Syrian Hamster IgG (BioXCell #BE0087) + Rat IgG2a (BioXCell #BE0089)). Mice were harvested at week 11 and spleen, lymph node and blood were isolated. All mice used in this study were housed in the Medical Sciences Research Building II Vivarium at Duke University Medical center in a pathogen-free environment with 12-h light/datk cycles at 20–25 ˚C, in accordance with Duke University Institutional Animal Care and Use Committee-approved animal protocols. All aspects of the procurement, conditioning/quarantine, housing, colony management, veterinary care, and carcass disposal programs comply with the guidelines set forth by the NIH guide for the care and use of laboratory animals, the Animal Welfare Act, and all applicable federal, state and institutional laws.

**Mouse phenotyping.** For mouse GCB and Tfh surface phenotype analysis, LN cells were resuspended in PBS containing 0.5% bovine serum albumin, 0.1% sodium azide and 1 mM EDTA (FACS buffer). Resuspended cells were first blocked with rat anti-mouse CD16/32 (2.4G2, BD, 1/200) and rat serum IgG (Sigma-Aldrich, 1/200) in FACS buffer for 30 min and stained with fluorochrome-conjugated antibodies specific for CD4 (GK1.5,BioLegend, 1/200), CD25 (PC61, BioLegend, 1/200), CD38 (90, BioLegend, 1/200), CD44 (IM7, BioLegend, 1/400), CD62L (MEL-14, BD, 1/200), CD69 (H1.2F3, BioLegend, 1/200), CD80 (16-10A1, BD, 1/100), CD138 (281-2, BD, 1/400), B220 (RA3-6B2, BioLegend, 1/200), CXCR5 (2G8, BD, 1/100), GL7 (GL7, BD, 1/200), I-A/I-E (M5/114.15.2, BD, 1/400), ICOS (C398.4 A, BioLegend, 1/200), IgD (11-26 c.2a, BD, 1/200), IgM (II/4, eBioscience, 1/200), PD-1 (29 F.1A12, BioLegend, 1/200), PD-L2 (TY25, BioLegend, 1/100) and TCRβ (H57-597, BD, 1/200) at 4 ℃ for 30 min. LIVE/DEAD

Fixable Near-IR Dead Cell Stain Kit (Life Technologies, #L34976, 1/1000) was subsequently used to determine the viability of cells. After surface staining, cells were washed, fixed, permeablized and stained with antibodies specific for Bcl-6 (K112-91, BD, 1/100) and FoxP3 (MF23, BD, 1/200). Cells were then resuspended to FACS buffer and analyzed on LSRII or LSRFortessa cell analyzer (BD Biosciences). The data analysis was performed using FACSDiva (BD Biosciences), and FlowJo software (Tree Star, Inc.).

**High-throughput IgH and IgL repertoire sequencing.** For IgH, a modified version of HTGTS-Rep-Seq[48] was used to cover both VDJ and SHM pattern of the whole human VH1-2 sequence. Briefly, starting from 400 ng genomic DNA extracted from sorted splenic GC B cells, a biotinylated human JH2 bait primer that anneals downstream of the JH2 segment was used to linearly amplify all the VDJ junctions containing this J. The biotin-labeled single-stranded DNA amplification products were enriched with streptavidin C1 beads (Thermo Fisher Scientific) and barcoded for Illumina MiSeq 2 × 300-bp paired-end next generation sequencing using Nextera (Illumina) P5-I5 and P7-I7 indexed primers. IgBLAST and Cloanalyst software was used to assign V, D, J segments, determine if the rearrangement was productive and find SHMs in productive VH1-2 sequences and determine CDR3 lengths. For IgL, nested PCR was used to amplify the knocked-in pre-rearranged VJ junction and tagged for MiSeq 2 × 300-bp paired-end sequencing. "SY" 2 AA deletion was determined by a SHM custom pipeline as previously described[82]. Primer sequences are shown below.

    hJH2 biogctgcagaccccagatacct
    hJH2 red nestedtggacagagaagactgggagg
    SA271LC_Fw_1stttcctcctgctactctggct
    SA271LC_Rv_1stccaatctcttggatggtgacca
    I5-SA271LC_Fw_2ndtctttccctacacgacgctcttccgatctggtttgatttagattacatgggtg
    P7-I7-
SA271LC_Rv_2ndcaagcagaagacggcatacgagatcggtctcggcattcctgctgaaccgctcttcc-
gatctctgaactgactttaactcctaaca

**Single-cell RNA-seq.** Single-cell RNA-seq was performed as described[83]. Briefly, lymph node cellular suspensions were loaded on a GemCode Single-Cell instrument (10X Genomics, Pleasanton, CA) to generate single-cell beads in emulsion. Single-cell RNA-seq libraries were then prepared using GemCode Single Cell 3′ Gel bead and library kit (10X Genomics). Single-cell barcoded cDNA libraries were quantified by quantitative PCR (Kappa Biosystems, Wilmington, MA) and sequenced on an Illumina NextSeq 500 (San Diego CA). Read lengths were 26 bp for read 1, 8 bp i7 index, and 98 bp read 2. Cells were sequenced to greater than 50,000 reads per cell. The Cell Ranger Single Cell Software Suite was used to perform sample de-multiplexing, barcode processing and single-cell 3′ gene counting. Reads were aligned to mouse genome assembly, mm10. from the Ensemble (www.ensemble.org)[83,84]. Graph based cell clustering, dimensionality reduction, and data visualization were analyzed by the Seurat R package[85]. Cells that exhibited high transcript counts, > 0.1% mitochondrial transcripts were excluded from analysis. Differentially expressed transcripts were determined in the Seurat R package utilizing the Likelihood-ratio test for single cell gene expression statistical test[86]. Graphics were generated using the Seurat and ggplot2 R packages. Single cell RNAseq data shown are from 2 mice in each group in aggregate. Biological pathway analysis was performed using String-db and Ingenuity Pathway Analysis software (Qiagen).

**Quantification and statistical analysis.** The statistical analysis for this paper were performed using Graphpad Prism 7 and SAS 9.4 utilizing exact Wilcoxon tests for group comparisons. Due to the exploratory nature of this research and the small sample size, we are using an alpha level of 0.05 as a descriptive level for significance and have not made any adjustments to control for multiple testing.

**Reporting summary.** Further information on research design is available in the Nature Research Reporting Summary linked to this article.

## Data availability
The RNA-seq and single-cell RNA seq raw reads have been deposited in the SRA database under accession PRJNA556211. The VRC01 KI mouse immunoglobulin VDJ sequence reads have been deposited in the SRA database under accession PRJNA603102. Source data for Figs. 1B, 1C, 1F–I, 2A–D, 4B–I, 5B–D, 5E, 6A–F, and Supplementary Figs. 1A–C, 1E–H. 2A–D, and 6E are provided as a source data file.

## Code availability
Code and other processed file formats are available from the corresponding authors upon reasonable request.

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

## Acknowledgements

F.W.A. is a Howard Hughes Medical Institute Investigator. Huan Chen is a Fellow of the Leukemia and Lymphoma Society. Flow cytometry was performed in the Duke Human Vaccine Institute Flow Cytometry Shared Resource. The authors thank Dr. Thomas Waldman for anti-Tac antibody sequences. P.B. is a Jenner Investigator. This work was supported by the Center for HIV/AIDs Vaccine Immunology-Immunogen Discovery (CHAVI-ID; UMI-AI100645) and the Consortia for HIV/AIDS Vaccine Development (CHAVD; UM1-AI44371) grants from NIH/NIAID/DAIDS and by the Medical Research Council (grant number MR/K012037). The funders had no role in data collection and interpretation, or the decision to submit the work for publication.

## Author contributions

Experimental Design: T.B., A.E., R.A., J.R.M., G.K., F.W.A., and B.F.H.; Investigation and assays: T.B., M.K., C.H.Y., M.T., H.C., D.W.C., X.C., C.C., R.P., M. Barr, L.L.S., R.M.S., C.M.B., H.B.V., S.S., K.W., M.G.L., A.O., P.B., D.M., M.B., M.A.M., L.V., K.O.S., J.R.M., G.K., F.W.A., and B.F.H.; Supervision: T.B., M.G.L., P.B., D.M., M.B., M.A.M., L.V., K.O.S., J.R.M., G.K., F.W.A., and B.F.H.; Data analysis: T.B., M.K., C.H.Y., M.T., H.C., D.W.C., R.P., M.B., K.W., and B.F.H.; Wrote paper: T.B. and B.F.H. with all other authors contributed to editing the paper; Funding: B.F.H.

## Competing interests

The authors declare no competing interests.
