## [Peer Review File · Nature Communications]

Reviewers' Comments:

Reviewer #1:

Remarks to the Author:

In this study, Bradley and colleagues evaluate the utility of immuno-modulation to enhance the elicitation of bNABs against HIV-1. With the focus on germline, lineage-based, and structure-guided vaccine design, it is often ignored that the human immune system pushed beyond normal antibody development to fully mature bNABs. It is clear that to achieve the massive somatic mutation necessary to achieve bNAb activity, immune checkpoints and tolerization mechanisms have been relaxed or modulated in these subjects. Thus, in the absence of a breakthrough in driving high levels of antibody maturation, there is likely a theoretical threshold on the level of somatic mutation that can be achieved in healthy humans by vaccination. Here the authors test directly whether modulating immune checkpoints increases the elicitation of bNABs by vaccination. This concept represents a major looming roadblock for the field, and breakthroughs in this arena are likely to be of significant interest and impact. The study is well-written and the authors are experts in the field. However, although there are several tantalizing findings, the effect on functional antibody responses of using immuno-modulators was a mixed bag and it is unclear whether these findings will have a direct and immediate impact on vaccine development. Importantly, their results fail to show that they can achieve enhanced antibody maturation.

There are several strong points in this study that overall indicate that immunomodulation altered B and CD4+ T cell responses, and that they had some degree of effect on anti-Env antibody responses. The most convincing data are from the KI mouse studies. Immune modulation increased germinal center and memory B cell frequencies for both treatments, which are important parameters for maturing B cell responses. It also increased TFH frequency, whereas TFR frequency was dampened, which are both desirable outcomes. However, the effect on somatic mutation was modest, and OX40 had only a modest effect on antibody binding. OX40 shifted the overall populations averages of VH/VL mutational frequency within B cells higher, but importantly, it did not increase the level of mutation that was achieved (i.e., the maximal mutational frequency that was induced). Thus, while there were more B cells with more mutations, OX40 did not significantly tip the needle toward high levels of mutation or enhanced affinity. CTLA-4 inhibition treatment was actually detrimental to mutation frequency, indicating that it may not be useful during vaccination, despite some enhancement of anti-Env antibody responses. Therefore one of the most critical milestones for such treatments was not achieved.

The transcriptomic data are also very interesting, and support the idea that the immunomodulators are actually influencing pathways that are relevant to the enhanced induction of B cell responses. In the single cell population data, it appears that OX40 created a transcriptomic shift in the B cell populations. However, tying these findings to functional outcomes was difficult.

The macaque data are less convincing. Much of what the authors describe as better or worse antibody responses are actually trends and are not statistically supported (e.g., Fig 1B, 1F, among many others). The lack of statistical support for many of the differences was driven by the low animal numbers of outbred CMs in each group (overall the small animal group numbers and lack of repetition throughout is a weakness). There are clearly trends indicating that immunomodulation altered immunity, but the lack of statistical support for many of the different readouts diminishes impact and enthusiasm about these findings. The authors need to significantly temper their descriptions of non-statistically supported data, and alter their conclusions accordingly.

Overall, it is not clear that these treatments enhanced the elicitation of broadly neutralizing antibodies. The authors note that they did not see heterologous neutralization. The enhancements

from each treatment that were observed were either for antibody binding or for autologous neutralization. These treatments clearly affected aspects of the B and T cell responses, but it is unclear whether these results significantly drive the field forward.

Minor points:

- 1- Is AUC the most appropriate readout for ELISA data here? Why not endpoint titers or EC50? The risk with AUC is masking the actual magnitude and character of the responses, and it is difficult to benchmark these responses against other published studies.
- 2- The serum composite score is not adequately described and it is not clear how this is a valid measure of responses. This is especially important because the most robust statistical support for treatment differences for macaques are for this score.
- 3- There is no testing of polyclonal avidity. This has been done in many other studies in both chaotrope assays and Octet BLI. If the argument is that there are enhanced antibody responses and maturation, this readout would be highly relevant. Especially for the CM data in which there was not a quantitative readout of mutation.
- 4- The method for VDJ NGS sequencing is not well described, and it is noted that it will be described later, but a major finding of this paper is based off of this method.
- 5- B cell culture and cloning is described in the materials and methods, but does not appear to be part of this study.

Reviewer #2:

Remarks to the Author:

T. Bradley et al, report very interesting and original data on the modulation of responses to HIV vaccines. One of the major objective in the field is to develop vaccines or strategies capable to elicit broad neutralizing antibodies (bnAbs) against HIV. The team has made previously outstanding contributions in the field of HIV vaccines, experimental models and the design of new immunogens.

The rationale behind this new study is that immunoregulatory mechanisms are associated with bnAb development in HIV-1 infected individuals. Therefore, authors tested the hypothesis that the modification of the immunoregulatory environment using blocking antibodies targeting inhibitory molecules (PD-1, CTLA-4) or agonist Ab of stimulatory receptors (OX40) during the vaccination may relieve the regulatory mechanisms blocking the induction of clonal lineages of bnAbs.

Authors have tested this hypothesis using several different Env HIV immunogens in two experimental models (NHP and nAbs knock in mice that express precursors of CD4 binding site directed bnAbs VRC01) (VRC01 mice).

This is a very interesting study with clear implications in the understanding of the effect of immunoregulatory molecules in the induction of HIV vaccine responses. CTLA-4 and PD-1 Abs have been tested in different models with the aim to improve T cell immunity. In this study, analyses are focused on "B-cell vaccines". Experimental vaccine regimens and data are well described. These results are original and conclusions are well supported by the data presented.

The first part of the results is obtained in NHP models and showed that sequential Env immunization associated with the co-administration of either CTLA-4 + PD-1 Abs or CTLA-4 Ab alone developed higher magnitude of IgG anti-Env as compared to PD-1 Ab alone and control groups. Neutralizing anti-Tier 1, but not Tier 2, HIV are detected. In a second set of NHP experiment, using a native-like SOSIP Env trimer boosted with sequential variants co-administered with CTLA-4 Abs, authors showed a higher magnitude of anti-Env IgG response and detection of anti-Tier 2 HIV antibodies in ¾ animals.

Transcriptomic analyses of sorted CD4+ T cells and B cells was performed and showed genes differentially expressed. Lymph node (LN) biopsies did not show significant changes in cell populations GC B cells, Tfh, Tfr cells in CTLA-4 treated animals.

Mice VRC01 experiments showed that CTLA-4 Ab and OX40 Ab groups had higher average serum antibody titers against all vaccine immunogens when compared to controls. GC and memory B cells were increased in LN from animals treated with CTLA-4 Abs. Both CTLA-4 and OX-40 Ab-treated groups had an increase of Tfh and a decrease of Tfr as compared to control group. Ig repertoire analysis showed contrasting results since OX-40 treatment increased somatic hypermutation frequency of the human IGHV1-2 bnAb precursor gene whereas this VH gene usage was decreased in CTLA-4 treated animals where an increased usage of endogenous mouse heavy chain genes was observed. Single cell RNAseq analysis of LN cells gave consistent results showing an upregulation of genes of cellular activation and down regulation of pro apoptotic genes in both CTLA-4 and OX-40 treated animals.

Comments:

Fig 1D: CTLA-4 and CTLA-4+PD-1 arms are grouped. This is not the usual way to present data in other figures showing Ab responses. Is the reason is statistical power?

Fig 2B right: it is not obvious what is the analysis presented here. Please clarify

Fig 3D: The level of Tfr is identical in animals treated with PD-1 and CTLA-4+PD-1 abs. Similarly, the % of Ki67+Tfh cells is the same in these groups and higher than in the control group. How the authors explain that these groups differ in terms of magnitude and neutralization IgG responses? Please comment.

Page 9: first paragraph. Authors should temper the conclusion that Notch and CSF may promote B cell response. Several studies have shown that Delta-1 Ligand of Notch may also activate regulatory T cells through the Notch pathway.

Fig 3: The data of Tfh KI67+ are interesting. However, in Fig 3G, the frequency of total Tfh seems to be lower in CTLA-4 treated animals as compared to controls except for one. Did the authors make the conclusion that the difference in term of IgG response is mainly due to cycling Tfh? Did they look at other markers and at the frequency of Env-specific Tfh cells in lymph nodes? Regarding the lower frequency of Tfh cells in Ab-treated animals (except one), this will be a good marker helping to demonstrate the role of these immunoregulatory molecules in the induction of B cell activation.

P10, first paragraph: "with all immune checkpoint Abs resulted in higher proportions of activated T cells after immunization." Authors should temper the conclusion of this paragraph (figure 3) since Ki67 was the only marker of activation presented and the difference between groups "is missing the significance threshold".

Fig 4B and 4C: Figures show binding response to sequential proteins (Fig4B) and comparison of the binding between 426c and heterologous SOSIP BG505 (Fig 4C). The magnitude of the response to 424c SOSIP is several logs below the magnitude of the response to other proteins (Fig 4B). This is the case in all groups. Please comment.

In the legend of Fig 4 it is noted n=2 in the control group while there are 4 animals in the control group (red squares).

Fig 5: results are not easy to interpret.

- i) The frequency of IGHV1-2*02 usage is lower in Ab-treated animals as compared to control group. This contradicts the initial hypothesis that the use of immunoregulatory antibodies is associated with increased clonal response. Please comment.
- ii) Moreover, CTLA-4 treated group, the response involved endogenous mice VH genes. It is not clear why OX40 treatment did not do the same?
- iii) in Fig 5F, the level of mutation in IGHV1-2*02 sequences is lower in CTLA-4 Ab-treated group as compared to the control group. This is in contradiction with the induction by this treatment group of GC B cells, memory B cells, Tfh cells and a decrease of Tfr cells (Fig 5A-D).

Fig 6E and F: Did the authors analyzed changes in gene expression in B cells in Ab-treated mice in comparison of B cells from control mice?

Response to reviewers' comments:

Reviewer #1:

1. There are several strong points in this study that overall indicate that immunomodulation altered B and CD4+ T cell responses, and that they had some degree of effect on anti-Env antibody responses. The most convincing data are from the KI mouse studies. Immune modulation increased germinal center and memory B cell frequencies for both treatments, which are important parameters for maturing B cell responses. It also increased TFH frequency, whereas TFR frequency was dampened, which are both desirable outcomes.

AUTHOR RESPONSE: We thank the reviewer for the encouraging comments and agree that demonstration that the germinal center can be modulated to improve antibody responses is an important outcome.

2. However, the effect on somatic mutation was modest, and OX40 had only a modest effect on antibody binding. OX40 shifted the overall populations averages of VH/VL mutational frequency within B cells higher, but importantly, it did not increase the level of mutation that was achieved (i.e., the maximal mutational frequency that was induced). Thus, while there were more B cells with more mutations, OX40 did not significantly tip the needle toward high levels of mutation or enhanced affinity. CTLA-4 inhibition treatment was actually detrimental to mutation frequency, indicating that it may not be useful during vaccination, despite some enhancement of anti-Env antibody responses. Therefore one of the most critical milestones for such treatments was not achieved.

AUTHOR RESPONSE: We agree with the reviewer that the effect on somatic mutation of the antibody genes was modest. OX40 and CTLA-4 are engineered to target T cell populations and while they may change the magnitude and frequency of the germinal center responses they may not have a direct effect on AID or the B cell SHM rates. Thus, novel immunotherapies specifically targeted B cell functions may be required to achieve higher SHM. We have included this important point in the discussion on page 18.

3. The transcriptomic data are also very interesting, and support the idea that the immunomodulators are actually influencing pathways that are relevant to the enhanced induction of B cell responses. In the single cell population data, it appears that OX40 created a transcriptomic shift in the B cell populations. However, tying these findings to functional outcomes was difficult.

AUTHOR RESPONSE: Although OX40 Ab perturbed the transcriptomic profiles more extensively than CTLA-4 Ab, they both had upregulation of transcripts important for cellular activation and proliferation which may have contributed to observations of increased germinal center Tfh and B cells and enhanced antibody titers observed in the mice. We have clarified this point at the end of the results on page 16.

4. The macaque data are less convincing. Much of what the authors describe as better or worse antibody responses are actually trends and are not statistically supported (e.g., Fig 1B, 1F, among

many others). The lack of statistical support for many of the differences was driven by the low animal numbers of outbreak CMs in each group (overall the small animal group numbers and lack of repetition throughout is a weakness). There are clearly trends indicating that immunomodulation altered immunity, but the lack of statistical support for many of the different readouts diminishes impact and enthusiasm about these findings. The authors need to significantly temper their descriptions of non-statistically supported data, and alter their conclusions accordingly.

AUTHOR RESPONSE: We agree with the reviewer that the individual macaque studies and individual immune assays when taken individually are only trends and not statistically significant due to the limited number of animals in each group. We have made sure to amend any language to clarify average increases or trends in the data that do not meet statistical significance. However, we believe the observation that CTLA-4 blocking Ab increased antibody responses can be concluded when examining the sum of the data presented in the manuscript that includes two independent macaque studies with multiple immune measures that demonstrate trends for increased HIV-1 antibody responses associated with CTLA-4 Ab. Nonetheless, we have indicated throughout the manuscript that only when assays are aggregated do we see statistical significance.

5. Overall, it is not clear that these treatments enhanced the elicitation of broadly neutralizing antibodies. The authors note that they did not see heterologous neutralization. The enhancements from each treatment that were observed were either for antibody binding or for autologous neutralization. These treatments clearly affected aspects of the B and T cell responses, but it is unclear whether these results significantly drive the field forward.

AUTHOR RESPONSE: We agree that we did not enhance the elicitation of bnAbs using our immunization strategies with CTLA-4 and OX40. We believe that several additional vaccine components will be required to generate bnAbs by a vaccine that includes, 1) optimizing the HIV-1 Env immunogen (Trimer, multimer), 2) mode of immunization (protein, DNA, mRNA) and 3) immunization timing and duration. We suggest in this manuscript that one additional factor for the field to consider is the addition of immune modulators to enhance the magnitude and durability of the neutralizing antibody responses. We have now stated these considerations on page 18 and 19 in the discussion.

6. Is AUC the most appropriate readout for ELISA data here? Why not endpoint titers or EC50? The risk with AUC is masking the actual magnitude and character of the responses, and it is difficult to benchmark these responses against other published studies.

AUTHOR RESPONSE: We believe that AUC gives of the most technically reliably measure of the data to determine changes in response and have used this measure in much of our previously published vaccine studies by ourselves and collaborators. The log Area Under the Curve (AUC) is the most reliable method to display binding or neutralization data, particularly when responses are low and the curves are not full sigmoid in shape, and this mode of data expression is used standardly by the HIV Vaccine Trials Network. A helpful reference here is from the HVTN

statistical unit led by Peter Gilbert is Yu, X, Gilbert, PB, Hioe CE, Zolla-Pazner, S and Self SG, Statistical Approaches to analyzing HIV-1 neutralizing antibody assay data. Stat. Biopharm. Res. 4: 1-13, 2012.

7. The serum composite score is not adequately described and it is not clear how this is a valid measure of responses. This is especially important because the most robust statistical support for treatment differences for macaques are for this score.

AUTHOR RESPONSE: The antibody composite score is an average of Z-scores that were converted for each individual immune measurement (binding, neutralization etc.) so that different types of assays could be averaged. We have added additional explanation to the results section on page 6 to describe this.

8. There is no testing of polyclonal avidity. This has been done in many other studies in both chaotrope assays and Octet BLI. If the argument is that there are enhanced antibody responses and maturation, this readout would be highly relevant. Especially for the CM data in which there was not a quantitative readout of mutation.

AUTHOR RESPONSE: We agree with the reviewer and think this is a critical experiment. We have isolated antibody IgG and performed binding affinity and avidity with SPR as per the reviewers suggestion. This is now included in the manuscript as figure 2. This assay demonstrated trends for higher average avidity measures for both of the macaque immunization studies for the CTLA-4 Ab treated animals. This may indicate a modest increase in antibody quality as well as quantity as observed in figure 1.

9. The method for VDJ NGS sequencing is not well described, and it is noted that it will be described later, but a major finding of this paper is based off of this method.

AUTHOR RESPONSE: We agree and this approach is a modification of the HTGTS-Rep-Seq protocol that has been published. We have modified the text to include more details of this approach in the methods.

10. B cell culture and cloning is described in the materials and methods, but does not appear to be part of this study.

AUTHOR RESPONSE: This section was included in error and we have removed it. We thank the reviewer for finding this error.

Reviewer #2:

1. T. Bradley et al, report very interesting and original data on the modulation of responses to HIV vaccines. One of the major objective in the field is to develop vaccines or strategies capable to elicit broad neutralizing antibodies (bnAbs) against HIV. The team has made previously outstanding contributions in the field of HIV vaccines, experimental models and the design of

new immunogens.

This is a very interesting study with clear implications in the understanding of the effect of immunoregulatory molecules in the induction of HIV vaccine responses. CTLA-4 and PD-1 Abs have been tested in different models with the aim to improve T cell immunity. In this study, analyses are focused on “B-cell vaccines”. Experimental vaccine regimens and data are well described. These results are original and conclusions are well supported by the data presented.

AUTHOR RESPONSE: We thank the reviewer for their positive remarks on our study and agree with their points.

2. Fig 1D: CTLA-4 and CTLA-4+PD-1 arms are grouped. This is not the usual way to present data in other figures showing Ab responses. Is the reason is statistical power?

AUTHOR RESPONSE: We observed that CTLA-4 and CTLA-4 + PD-1 Ab treated animals had a similar increase in magnitude of HIV Env antibody responses so we grouped them for statistical power. We have now noted this in the main text on page 6.

3. Fig 2B right: it is not obvious what is the analysis presented here. Please clarify

AUTHOR RESPONSE: We have now clarified the figure legend (now 3B) to show that these are selected genes that have previously shown to play a role in the regulation of CD4 T cell proliferation and activation. The red highlighted genes are ones that were changed in both the CTLA-4 and CTLA-4 plus PD-1 groups.

4. Fig 3D: The level of Tfr is identical in animals treated with PD-1 and CTLA-4+PD-1 abs. Similarly, the % of Ki67+Tfh cells is the same in these groups and higher than in the control group. How the authors explain that these groups differ in terms of magnitude and neutralization IgG responses? Please comment.

AUTHOR RESPONSE: We are measuring total frequency of Tfr and Tfh that are changed by administration of CTLA-4 and PD-1 blocking Abs and not only HIV-specific responses. This observation indicates that while PD-1 Ab increased global Tfr and all Ab treatments increased global Ki67+ Tfh in the lymph node this was not the sole driver of the changes in blood antibody titers that we observed. Likely other regulatory and enhancing cell types were important for this effect or the HIV-1-specific T cell response was masked by these global changes. We believe this data is important to present because it demonstrates changes in T cell types in lymphoid tissue in healthy animals during immunotherapeutic Ab administration that should be considered for human trials investigating use of these drugs.

5. Page 9: first paragraph. Authors should temper the conclusion that Notch and CSF may promote B cell response. Several studies have shown that Delta-1 Ligand of Notch may also activate regulatory T cells through the Notch pathway.

AUTHOR RESPONSE: We agree and have now noted this on page 10 of the text.

6. Fig 3: The data of Tfh KI67+ are interesting. However, in Fig 3G, the frequency of total Tfh

seems to be lower in CTLA-4 treated animals as compared to controls except for one. Did the authors make the conclusion that the difference in term of IgG response is mainly due to cycling Tfh? Did they look at other markers and at the frequency of Env-specific Tfh cells in lymph nodes? Regarding the lower frequency of Tfh cells in Ab-treated animals (except one), this will be a good marker helping to demonstrate the role of these immunoregulatory molecules in the induction of B cell activation.

AUTHOR RESPONSE: There is a modest reduction in the percentage of CD4 Tfh in the second macaque study but not the first if we excluded the animal with the high frequency. We would like to note that the magnitude of this change (from an average of ~3% to ~1.8%) is unclear what the biological significance would be. We believe that further study with additional macaques will be required to determine if this is a biologically relevant change that affects B cell responses. We were not able to study the Env-specific Tfh cells in the lymph node, but agree with the reviewer that this will be critical for evaluating these immunoregulatory molecules in future studies. We have now stated this on page 17 of the discussion section.

7. P10, first paragraph: "with all immune checkpoint Abs resulted in higher proportions of activated T cells after immunization." Authors should temper the conclusion of this paragraph (figure 3) since Ki67 was the only marker of activation presented and the difference between groups "is missing the significance threshold".

AUTHOR RESPONSE: We agree and modified this sentence in the results text now on page 11.

8. Fig 4B and 4C: Figures show binding response to sequential proteins (Fig4B) and comparison of the binding between 426c and heterologous SOSIP BG505 (Fig 4C). The magnitude of the response to 424c SOSIP is several logs below the magnitude of the response to other proteins (Fig 4B). This is the case in all groups. Please comment.

AUTHOR RESPONSE: The SOSIP trimers represent near-native trimer structures and more difficult to induce binding antibodies to in animals as these proteins have a fully intact glycan shield. This is similar to what was observed in previous studies (Tian et al. *Cell*. 2016). Moreover, plasma antibody binding was measured at the conclusion of immunization where animals only received 1 immunization of the SOSIP trimer compared to multiple immunization with non-native proteins to guide B cell development. We have included comment on the magnitude in the results section on page 13

9. In the legend of Fig 4 it is noted n=2 in the control group while there are 4 animals in the control group (red squares).

AUTHOR RESPONSE: There were indeed 4 animals in the control group that consisted of 2 animals from that were treated with isotype control antibody and 2 animals treated with PBS. We thank the reviewer for identifying this error and have corrected the legend to show 4 animals in the control group and stated more clearly the definition of the control group on page 13 of the results.

10. Fig 5: results are not easy to interpret.

- i) The frequency of IGHV1-2*02 usage is lower in Ab-treated animals as compared to control group. This contradicts the initial hypothesis that the use of immunoregulatory antibodies is associated with increased clonal response. Please comment.
- ii) Moreover, CTLA-4 treated group, the response involved endogenous mice VH genes. It is not clear why OX40 treatment did not do the same?
- iii) in Fig 5F, the level of mutation in IGHV1-2*02 sequences is lower in CTLA-4 Ab-treated group as compared to the control group. This is in contradiction with the induction by this treatment group of GC B cells, memory B cells, Tfh cells and a decrease of Tfr cells (Fig 5A-D).

AUTHOR RESPONSE: i) The frequency of the human IGHV1-2*02 was lower indicating increased usage of endogenous mouse genes. We believe this demonstrates diversification of the germinal center response by allowing B cells that did not normally participate to be included. Likely due to decreased immunoregulation. ii) These molecules are targeting different receptors that are present on different populations of T cells and thus may alter the germinal center in different ways. OX40 that is on Tfh cells may have had a more dramatic effect on providing help to B cells and increasing SHM, whereas CTLA-4 Ab did not have this effect. iii) We believe this data demonstrates that while there may have been a higher frequency of GC B cells with CTLA-4 Ab treatment, that many of them did not have an increase in SHM, but with OX40 Ab there was an increase in both SHM and cell frequency. The observation that CTLA-4 Ab B cells did have higher SHM suggests that immunomodulatory agents that are developed to target B cell SHM may be needed beyond OX40. We have now indicated these points in the discussion.

11. Fig 6E and F: Did the authors analyzed changes in gene expression in B cells in Ab-treated mice in comparison of B cells from control mice?

AUTHOR RESPONSE: We did not isolate B cells and perform RNA-sequencing, but we did subset B cells by expression of B cell-specific transcripts (*Cd79a*) and analyzed gene expression changes only in B cells. These results are in figure 7F. We have now included the differentially expressed gene lists for these B cells as supplementary tables 9 and 10.

Reviewers' Comments:

Reviewer #1:

Remarks to the Author:

The revised manuscript is improved over the previous version, and the authors were responsive to most criticisms by including revisions in the text. The authors have also added data regarding antibody maturation in the form of SPR experiments to assess trends in binding avidity. Importantly, several critical points have now been more thoroughly explained, including how certain analyses were done and how certain readouts are obtained, and the language surrounding statistics has been tempered.

Reviewer #2:

Remarks to the Author:

The authors answered the questions perfectly. The new results and comments provided are entirely satisfactory.

REVIEWERS' COMMENTS:

Reviewer #1 (Remarks to the Author):

The revised manuscript is improved over the previous version, and the authors were responsive to most criticisms by including revisions in the text. The authors have also added data regarding antibody maturation in the form of SPR experiments to assess trends in binding avidity. Importantly, several critical points have now been more thoroughly explained, including how certain analyses were done and how certain readouts are obtained, and the language surrounding statistics has been tempered.

We thank the reviewer for their time and comments.

Reviewer #2 (Remarks to the Author):

The authors answered the questions perfectly. The new results and comments provided are entirely satisfactory.

We thank the reviewer for their time and comments.